# CraftRTL: High-quality Synthetic Data Generation for Verilog Code Models with Correct-by-Construction Non-Textual Representations and Targeted Code Repair

**Mingjie Liu**[*], **Yun-Da Tsai**[*], **Wenfei Zhou, Haoxing Ren**
NVIDIA Corporation
`{mingjiel, yundat, wenfeiz, haoxingr}@nvidia.com`

## Abstract

Despite the significant progress made in code generation with large language models, challenges persist, especially with hardware description languages such as Verilog. This paper first presents an analysis of fine-tuned LLMs on Verilog coding, with synthetic data from prior methods. We identify two main issues: difficulties in handling non-textual representations (Karnaugh maps, state-transition diagrams and waveforms) and significant variability during training with models randomly making "minor" mistakes. To address these limitations, we enhance data curation by creating correct-by-construction data targeting non-textual representations. Additionally, we introduce an automated framework that generates error reports from various model checkpoints and injects these errors into open-source code to create targeted code repair data. Our fine-tuned Starcoder2-15B outperforms prior state-of-the-art results by 3.8%, 10.9%, 6.6% for pass@1 on VerilogEval-Machine, VerilogEval-Human, and RTLLM.

## 1 Introduction

Large Language Models (LLMs) have achieved significant success across various natural language processing tasks and have extended their capabilities to code generation, leading to the development of specialized models targeting code generation. The effectiveness of these models is largely influenced by the size and quality of their training datasets, as highlighted by scaling laws (Achiam et al., 2023; Zhang et al., 2024a). Prominent code LLMs have set new benchmarks records by utilizing extensive, synthetically generated datasets through methods like Self-Instruct (Wang et al., 2022; Chaudhary, 2023), Evol-Instruct (Xu et al., 2023), and OSS-Instruct (Wei et al., 2023). These synthetic data generation techniques allow code LLMs to generate a wide range of complex code examples, enhancing their training and performance in real-world coding scenarios.

While most code LLMs concentrate on software programming languages, there is increasing interest in developing models for hardware description languages (HDLs), which are essential for chip design and hardware verification. Despite efforts to collect and synthesize more diverse Verilog code to enhance specialized code LLMs (Liu et al., 2023c; Pei et al., 2024; Cui et al., 2024; Zhao et al., 2024), HDLs still face challenges akin to those encountered in low-resource languages (Cassano et al., 2022). These challenges are mainly due to the limited availability of high-quality instruction-following data and the constrained capability of existing LLMs to generate RTL code, which affects the models' performance and their ability to generalize across programming languages.

Developing high-quality synthetic Verilog code for training code large language models (LLMs) faces significant challenges due to two primary factors. Firstly, Verilog is considered a low-resource language (Cassano et al., 2022), meaning there is a scarcity of available training data compared to high-resource software programming languages like Python. This limited data availability restricts the models' ability to learn diverse and complex coding patterns effectively. Secondly, verifying the correctness of hardware description language (HDL) code, such as Verilog, is inherently more

---

[*]Equal Contributions.

complex than verifying software code. While software code correctness can often be assessed using random test cases and automated unit tests (Chen et al., 2022), hardware code requires comprehensive testbenches and rigorous verification planning and methodologies. This additional complexity makes it challenging to ensure that synthetic Verilog code is functionally accurate (Bhandari et al., 2024; Qiu et al., 2024), posing a barrier to improving model performance.

In this paper, we start with a thorough analysis of fine-tuned large language models (LLMs) applied to Verilog code, using synthetic data techniques from previous works. Our analysis reveals two key issues: (1) models have difficulty handling non-textual elements in problem statements, indicating challenges in interpreting complex or unconventional inputs; and (2) there is notable variability in the models' pass rates across different benchmark problems and training checkpoints, exposing inconsistencies in learning outcomes, often due to the models making "minor" programming mistakes.

Given the limitations identified in our analysis of relying solely on LLMs for generating synthetic data, we shift our focus to improving data curation to address these issues. Current LLMs frequently struggle with interpreting and processing non-textual representations and are insufficient in generating effective testbenches for evaluating solution quality. Therefore, instead of depending exclusively on LLMs to address data quality concerns, we develop targeted fine-tuning data to better mitigate these problems. Experimental results demonstrate that our models achieve state-of-the-art (SOTA) results on VerilogEval (Liu et al., 2023b) and RTLLM v1.1 (Lu et al., 2024) benchmarks, outperforming prior works by large margins on problems with human-level description. The major contributions of this paper are as follows:

- We perform a thorough analysis of fine-tuned LLMs on Verilog code using previously established synthetic data generation methods, uncovering challenges with non-textual elements and notable variability in performance across benchmark problems during training.

- We create correct-by-construction data to ensure solution correctness, incorporating Karnaugh Maps, state-transition diagrams, and waveforms, which significantly enhance the model's ability to handle non-textual representations.

- We develop an automated framework that utilizes LLMs to generate error reports from benchmark problems at various checkpoints, which are then injected into open-source code to create a fine-tuning dataset targeted at correcting the model's specific "minor" mistakes.

- We rigorously evaluate the latest foundational and frontier code models. We note that recent advanced models like GPT-4o already reached competitive performance compared to previous efforts targeting Verilog code generation.

- Experimental results demonstrate that models fine-tuned with our data achieve state-of-the-art performance on Verilog coding. Specifically, our fine-tuned model based on Starcoder2-15B (Lozhkov et al., 2024) outperforms prior SOTA results by 3.8%, 10.9%, 6.6% for pass@1 on VerilogEval-Machine, VerilogEval-Human, and RTLLM, respectively.

## 2 EXAMINING FINE-TUNED LLMS USING SYNTHETIC GENERATED DATA ON VERILOG CODING

In this section, we start with a thorough analysis of fine-tuned large language models (LLMs) applied to Verilog code. We adapt previous approaches for generating synthetic data for general coding to focus on Verilog code. For our pilot study, we only present results based on fine-tuning StarCoder2-15B (Lozhkov et al., 2024). Details on experimental settings are the same as in Section 4. We assess model performance in Verilog code completion and identify two main issues. First, the models demonstrate notably poor performance when dealing with non-textual elements in the problem statements. Second, the variability in the models' pass rates across different benchmark problems and training checkpoints suggests inconsistencies in learning outcomes and model variability.

### 2.1 SYNTHETIC DATA GENERATION FOR VERILOG CODING

We build on previous methods for synthetic data generation by applying Self-Instruct (Wang et al., 2022) and OSS-Instruct (Wei et al., 2023) with custom prompt templates tailored for Verilog coding. To enhance data coverage and diversity, we supplement these techniques with additional context

from Wikipedia and textbooks. We also prompt models to generate problem descriptions to include non-textual representations.

We use *nemotron-4-340b-instruct* (Nvidia et al., 2024) selected for its open license that allows commercial use. Our process includes deduplication and a decontamination procedure akin to that outlined by Li et al. (2023). Additionally, we conduct syntax checks to eliminate coding problems containing docstrings or solutions from Verilog benchmarks. To ensure further data quality, we discard code solutions that fail these syntax checks and apply self-verification (Weng et al., 2023) to remove entries where the LLM identifies errors in the solution. Table 1 shows the quantity of our synthetic data generation (denoted as SDG) after deduplication and filtering, yielding a total of 80.1k fine-tuning examples.

Table 1: Data quantity SDG.

| Method | Quantity |
| --- | --- |
| Self-Instruct | 24.7k |
| OSS-Instruct | 28.4k |
| Docu-Instruct | 12.0k |
| Non-textual | 15.0k |
| SDG Total | 80.1k |

**Self-Instruct**   We follow the approach outlined in Wang et al. (2022) to generate synthetic Verilog coding problems. Initially, we randomly generate from the LLM and curate 50 questions that request Verilog coding problems without any in-context examples. From these, we then randomly choose 1 to 5 seed questions to use as in-context examples.

**OSS-Instruct**   We begin by processing pretraining code data to extract our seed code from *The Stack v2* (Lozhkov et al., 2024), focusing on Verilog and SystemVerilog. Following the approach in Liu et al. (2023b), we post-process this data by selecting self-contained Verilog code that passes syntax checks using Pyverilog (Takamaeda-Yamazaki, 2015). With the refined seed code data, we then prompt large language models (LLMs) to use this code as inspiration for generating Verilog coding problems similar to Wei et al. (2023).

**Docu-Instruct**   Drawing inspiration from Nvidia et al. (2024) and Sudalairaj et al. (2024), we utilize document sources from Wikipedia and textbooks for instruction generation. We begin by filtering Wikipedia entries, prompting the LLM to classify whether the content pertains to hardware design or Verilog coding concepts. Additionally, we manually selected approximately relevant 100 textbooks. These textbooks are then segmented into chunks of paragraphs or sentences, ensuring each chunk contains fewer than 2k tokens.

**Non-textual Representations**   VerilogEval-Human (Liu et al., 2023b) includes benchmark problems involving non-textual representations. For example, Boolean logic tables and Karnaugh maps are presented in tabular formats, state-transition diagrams for finite state machines are depicted as edge lists and sequential waveforms are described in tables with signal outputs recorded at various time steps. To incorporate such representations, we encouraged LLMs to generate problems from open-source code, with instructions to utilize these tabular data structures.

## 2.2 Challenges with Non-Textual Representations

We observe that models underperform on benchmark problems involving non-textual input formats, such as Karnaugh Maps, state-transition diagrams, and waveforms. Table 2 shows the pass@1 results for the VerilogEval (Liu et al., 2023b). Additionally, we have identified a subset of 45 questions within VerilogEval-Human that include non-textual representations, termed VerilogEval-NonText. It appears that models like GPT-4o and Starcoder2 strug-

Table 2: pass@1 results on VerilogEval sampled with temperature of 0.8.

| Model | Machine | Human | NonText |
| --- | --- | --- | --- |
| GPT-4o | 63.7 | 55.4 | 27.0 |
| Starcoder2 | 57.7 | 29.1 | 10.3 |
| Starcoder2-SDG | 73.7 | 47.4 | 22.2 |

gle with these non-textual formats, likely due to insufficient representation of such data during both pretraining and fine-tuning. Despite our efforts to generate such questions during synthetic data creation, our fine-tuned models still lag in these areas. This outcome is not entirely surprising, given that the LLMs used were also ineffective at generating problems with these representations, complicating the validation of fine-tuning data. These results suggest that merely including non-textual data is insufficient; ensuring the quality and correctness of the data, particularly that the code solutions accurately align with these representations, is crucial.

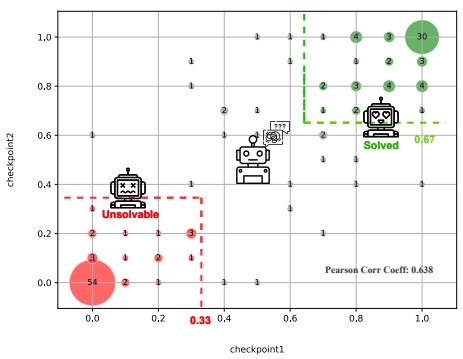 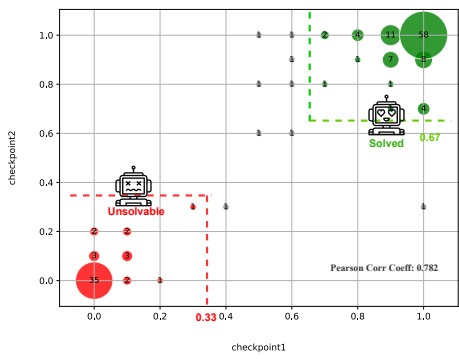

(a) Starcoder2-15B on SDG.
(b) Starcoder2-15B on SDG-CC-Repair.

Figure 1: Our methods reduce pass rate variability during training: SDG (left) shows high volatility with significant degradation on many problems, while SDG-CC-Repair (right) stabilizes learning outcomes on solvable problems (details in Appendix A.10).

## 2.3 VARIABILITY ON PASS RATES DURING TRAINING

During our training, we observed significant variability in the model's pass rate on specific benchmark problems across different checkpoints. We note such variance is different from training instability (Wortsman et al., 2023) as we observe a stable decrease in the training loss. This variability persists even in the later stages of training, despite using a low learning rate. We illustrate this variability in Figure 1a. The scatter plot tracks the pass rate for each problem in VerilogEval-Human, with each point representing the pass rate for the same problem across two checkpoints. The size of each point indicates the number of problems with the same pass rates for the two model checkpoints. We further categorize the region into areas where the checkpoints agree on problem difficulty and areas where they do not.

Alarmingly, we find that nearly 15% of the problems show significant discrepancies between these two checkpoints, with an equal number of problems demonstrating improvement and degradation. Our detailed analysis of the sampled code completions for such problems when pass rate degrades suggests that the model is generally on the right track but makes "minor" errors that are small, detailed, and seemingly trivial. While it is possible that LLMs experience catastrophic forgetting during fine-tuning (Luo et al., 2024a), we do not anticipate this being a major factor due to the low learning rate and the small number of gradient updates (64 steps with 16k data samples). Instead, we believe the primary issue is our inability to ensure the quality of our data, particularly in verifying whether the sampled code solutions correctly solve the code problems.

## 3 IMPROVING VERILOG CODING WITH CORRECT-BY-CONSTRUCTION NON-TEXTUAL REPRESENTATIONS AND TARGETED CODE REPAIR

Based on our detailed analysis of the limitations of relying solely on LLMs for generating synthetic data, we focus our data curation efforts to address these shortcomings. Our goal is to enhance data quality and ensure the correctness of solutions for the generated problems. We have found that current LLMs often lack the capability to understand and process non-textual representations effectively and are unable to generate satisfactory testbenches for assessing solution quality. Consequently, rather than depending entirely on LLMs to resolve data quality issues, we instead create targeted fine-tuning data to mitigate these problems.

## 3.1 ENSURING QUALITY THROUGH CORRECT-BY-CONSTRUCTION

We generate Verilog code problems and solutions that are correct-by-construction. Our focus is on creating problems and solutions for non-textual representations. Table 3 shows the quantity of our correct-by-construction generation data (referred to as CC). To prevent data contamination, we exclude entries that duplicate the data representations of benchmark problems.

**Karnaugh Maps and Truth Tables (KMap)**    We start by sampling random configurations, which include selecting the number of variables and their names. After determining the number of variables, we randomly choose valid minterms and don't-cares. For $n$ variables, there are $2^n$ possible states, and each state can be assigned one of three values (0, 1, or x), leading to $3^{2^n}$ possible combinations of minterms and don't-cares. From these minterms, we derive the sum-of-products (SOP) form to represent the Boolean logic. We then create Truth Tables and Karnaugh Maps based on the chosen minterms and don't-cares. In the KMap, Gray encoding is used as default for the row and column sequences to ensure that only a single bit changes between adjacent cells. Additionally, we apply modifications by transposing the map and randomly swapping adjacent rows or columns. We randomly sample from $n = \{3, 4\}$ variables.

Table 3: Data quantity CC.

| Method | Quantity |
|---|---|
| KMap | 12.5k |
| FSM | 8.0k |
| Waveforms | 8.0k |
| CC Total | 28.5k |

**State Transition Graphs and Tables (FSM)**    We construct problems for finite-state machines (FSMs) with state-transition representations with a similar approach to KMaps. We begin by sampling random configurations, including the number of states (e.g., 4, 6, or 10) and the bit width of the input (e.g., 1 or 2). We then create the transition graph, ensuring that it is both meaningful and legally defined. We generate state-transition graphs for both Moore and Mealy state machines. From these graphs, we produce edge-list and transition table representations. Finally, we construct the Verilog code to implement the logic for state transitions and output assignments.

Algorithm 1 outlines the process for generating a Moore FSM with random transitions. State reachability is ensured by first constructing a tree. Legality for state transition is ensured by ensuring each node has an out-degree of $2^w$ with the input bit width of $w$. The result is an FSM where transitions between states are randomly assigned but conform to the specified input bit width. The algorithm can be easily modified for a Mealy FSM by assigning the output to the edges rather than nodes.

---
**Algorithm 1** Generate transition graph for Moore FSM.

---
**Input:** Number of states $n$, bit width of input $w$
**Output:** FSM graph with transitions and states
Initialize the number of states $n$ and bit width of input $w$
Randomly generate a tree with $n$ nodes
Define the root of the tree as the reset state
**for** each node in the tree **do**
    Assign a unique state to the node
    Assign an output to the node
**end for**
**for** each node in the tree **do**
    Add additional transition edges to form a graph
    Ensure that each node has an out-degree of $2^w$
**end for**

---

Figure 2 illustrates our approach for generating state transition logic in Verilog from a state-transition graph. Our method predominantly employs an out-edge focused strategy for state transitions. Additionally, we incorporate in-edge focused transition logic to address specific challenges encountered in benchmark problems. These benchmarks often involve states represented using one-hot encoding and require rigorous testing of non-default states.

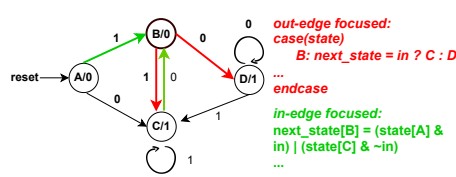

Figure 2: State transition logic.

**Waveforms**    We utilize correct-by-construction code solutions for both KMaps and FSMs. Because these codes are generated using similar templates, designing corresponding testbenches is straightforward. We simulate the generated code to produce waveform Value Change Dump (VCD) files. These VCD files are then parsed and converted into waveform representations. Our approach covers KMaps as combinational circuits and FSMs as sequential circuit waveforms.

## 3.2    MITIGATING "MINOR" ERRORS WITH TARGETED CODE REPAIR

Our analysis revealed that the models were generally on the right track to correct solutions but were making minor errors—small, detailed, and seemingly trivial. Unlike complex, unsolvable problems, these minor errors could be easily corrected by language models. This insight led us

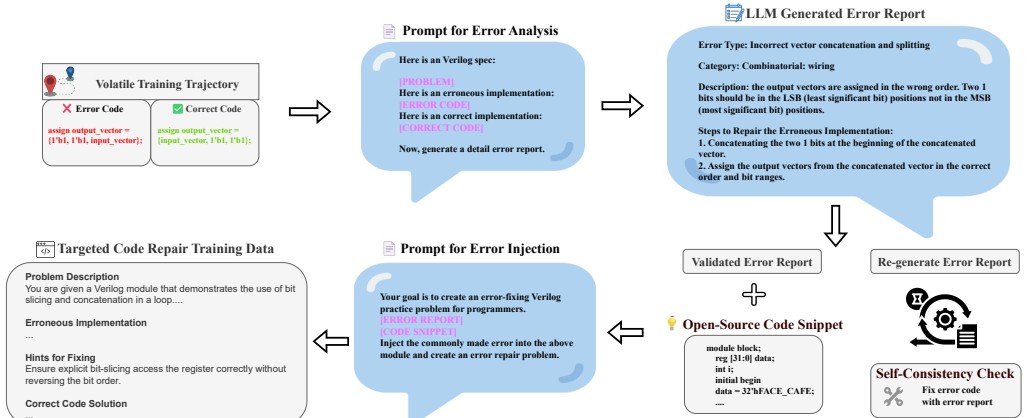

Figure 3: Overview of our approach for generating targeted code repair data: (1) prompting the LLM to generate detailed error reports from correct and erroneous code, (2) validating error report quality by ensuring the LLM can debug the errors based on the report, and (3) leveraging the LLM to inject similar errors into open-source code, creating a diverse training dataset.

to develop a new strategy centered on targeted error code repair. Our approach includes creating detailed error reports on benchmark problems, re-creating these errors on correct open-source code, and conducting rigorous validation to ensure quality. We use *nemotron-4-340b-instruct* as the LLM to construct our targeted **Repair** data. We generated 847 error reports across the three benchmarks and produced 2,736 data samples. After filtering, this resulted in a final set of 1,406 targeted code repair data points.

**Error Report Construction**    To systematically address the issue, we first created a comprehensive Error Report for benchmark problems using LLMs, targeting those with significant pass rate fluctuations across training checkpoints for models on SDG data. We prompt the LLM to examine the nature of the mistakes by comparing correct and erroneous code completions for each problem, categorizing the errors into common error types (details in Appendix A.9). This detailed report not only categorizes the errors but also highlights areas where the model consistently underperforms.

**Targeted Code Repair Dataset**    Building on the error report, we further develop a targeted code repair dataset to address these common errors. This dataset is constructed using two main sources: the errors identified in the Error Report and correct code snippets gathered from open-source repositories. We introduced the identified errors into correct code snippets to create repair problems, which include a problem description, erroneous code implementation, and hints about the nature of the error and how to fix it. This targeted strategy enables the model to learn how to avoid common errors and generate improved code completions, thereby enhancing model accuracy.

**Quality Assurance with LLM Validation**    To ensure the reliability of the error report and the code repair dataset, we implemented a two-phase validation process with LLMs. In the first phase, we conducted a self-consistency check of the Error Report by having the language model attempt to the fix error code based on the report's hints. This step verifies the accuracy of the report by confirming that the model can resolve the errors using the provided guidance, whereas directly prompting the LLM without detailed error reports could resolve only 13% of the errors. In the second phase, during the generation of the code repair dataset, we apply self-verification, including deduplication, syntax filtering, and benchmark decontamination. These measures ensure the dataset's quality and uniqueness, preventing overlap with evaluation benchmarks.

# 4    EXPERIMENT

## 4.1    IMPLEMENTATION DETAILS

**Training Data**    Our fine-tuning training data is comprised of 80.1k LLM synthetic generated data using various prompting methods as described in Section 2.1, 28.5k data samples generated correct-by-construction aimed at non-textual representations detailed in Section 3.1, and 1.4k carefully fil-

Table 4: We compare our models with various baseline models on VerilogEval (Liu et al., 2023b). We update the results from Zhao et al. (2024) with the latest foundational and frontier code models. The **best** results are highlighted in bold.

| Type | Model | Size | VerilogEval (Liu et al., 2023b) | | | | | |
|---|---|---|---|---|---|---|---|---|
| | | | Machine (%) | | | Human (%) | | |
| | | | pass@1 | pass@5 | pass@10 | pass@1 | pass@5 | pass@10 |
| Foundational Models | Llama-3.1 | 8B | 48.7 | 67.3 | 74.1 | 26.9 | 37.8 | 44.2 |
| | Llama-3.1 | 405B | 67.3 | 75.1 | 76.9 | 53.8 | 61.0 | 62.8 |
| | Nemotron-4 | 340B | 53.0 | 60.3 | 62.2 | 43.1 | 48.3 | 50.0 |
| | GPT-3.5-turbo | - | 58.0 | 74.0 | 77.6 | 31.2 | 44.1 | 47.4 |
| | GPT-4o | - | 65.9 | 71.4 | 72.7 | 57.1 | 63.9 | 66.7 |
| Code Models | CodeLlama | 7B | 43.1 | 47.1 | 47.7 | 18.2 | 22.7 | 24.3 |
| | CodeQwen | 7B | 46.5 | 54.9 | 56.4 | 22.5 | 26.1 | 28.0 |
| | Starcoder2 | 15B | 68.7 | 82.3 | 88.5 | 37.7 | 50.6 | 57.2 |
| | DeepSeek-Coder | 6.7B | 52.2 | 55.4 | 56.8 | 30.2 | 33.9 | 34.9 |
| | DeepSeek-Coder-V2 | 16B | 67.4 | 78.3 | 81.8 | 46.9 | 55.9 | 58.9 |
| | DeepSeek-Coder-V2 | 236B | 68.2 | 74.1 | 76.2 | 56.4 | 62.2 | 66.0 |
| RTLCoder (Liu et al., 2023c) | Mistral | 7B | 62.5 | 72.2 | 76.6 | 36.7 | 45.5 | 49.2 |
| | DeepSeek-Coder | 7B | 61.2 | 76.5 | 81.8 | 41.6 | 50.1 | 53.4 |
| BetterV (Pei et al., 2024) | CodeLlama | 7B | 64.2 | 75.4 | 79.1 | 40.9 | 50.0 | 53.3 |
| | DeepSeek-Coder | 6.7B | 67.8 | 79.1 | 84.0 | 45.9 | 53.3 | 57.6 |
| | CodeQwen | 7B | 68.1 | 79.4 | 84.5 | 46.1 | 53.7 | 58.2 |
| CodeV (Zhao et al., 2024) | CodeLlama | 7B | 78.1 | 86.0 | 88.5 | 45.2 | 59.5 | 63.8 |
| | DeepSeek-Coder | 6.7B | 77.9 | **88.6** | **90.7** | 52.7 | 62.5 | 67.3 |
| | CodeQwen | 7B | 77.6 | 88.2 | **90.7** | 53.2 | 65.1 | 68.5 |
| OriGen (Cui et al., 2024) | DeepSeek-Coder | 6.7B | 74.1 | 82.4 | 85.7 | 54.4 | 60.1 | 64.2 |
| Ours SDG-CC-Repair | CodeLlama | 7B | 78.1 | 85.5 | 87.8 | 63.1 | 67.8 | 69.7 |
| | DeepSeek-Coder | 6.7B | 77.8 | 85.5 | 88.1 | 65.4 | 70.0 | 72.1 |
| | Starcoder2 | 15B | **81.9** | 86.9 | 88.1 | **68.0** | **72.4** | **74.6** |

tered data for targeted code repair as outlined in Section 3.2. We refer to each data set as **SDG**, **CC**, and **Repair**, respectively.

**Pretrained Models**    Following prior work, we use CodeLlama-7b-Instruct (Roziere et al., 2023) and Deepseek-Coder-6.7b-Instruct (Guo et al., 2024) as the base model, formatting our data according to their default chat prompt templates. Additionally, we explore the Starcoder2-15B (Lozhkov et al., 2024) model in our experiments.

**Model Training**    Training is conducted with 32 NVIDIA A100-80GB GPUs through the Distributed Data Parallel (DDP) module from PyTorch. We set the learning rate at 5e-5 for CodeLlama and DeepSeek-Coder, and 1e-5 for Starcoder2. We use Adam (Kingma & Ba, 2017) as our optimizer with full parameter updates and truncate sequence lengths longer than 4096 tokens. We used a batch size of 256 samples. We fine-tune models for 1 epoch using a standard cross entropy loss on the response tokens (while masking loss on prompt tokens).

**Model Inference**    We use vLLM (Kwon et al., 2023) where the inference engine is set up with bf16 dtype, tensor parallel size of 8, and a maximum token limit of 4096. We sample each problem 20 times. We report the best results from two different temperatures 0.2 and 0.8, as consistent with prior work (Liu et al., 2023c; Zhao et al., 2024).

## 4.2 Evaluation Metric and Benchmark

**Evaluation Metric** Following prior work (Chen et al., 2021; Liu et al., 2023a), for each experiment we use the unbiased pass@k metric to measure the Verilog generation accuracy. The pass@k metric estimates the proportion of problems that can be solved at least once in k attempts:

$$pass@k := \mathbb{E}_{\text{Problems}} \left[ 1 - \frac{\binom{n-c}{k}}{\binom{n}{k}} \right], \tag{1}$$

where $n \geq k$ represents the total number of trials for each problem, and $c$ represents the number of trials that pass the functional check.

**VerilogEval** (Liu et al., 2023b) contains two subsets of problems, where VerilogEval-Human contains manually converted problem descriptions from the original HDLBits website, and VerilogEval-Machine with GPT-3.5 generated problem descriptions.

Table 5: Evaluations on RTLLM v1.1 (Lu et al., 2024) using unbiased pass@k metrics. The **best** results are highlighted in bold. We re-evaluate all models (see Appendix A for details).

| Type | Model | Size | RTLLM v1.1 (Lu et al., 2024) | | | | | |
| | | | Syntax (%) | | | Func. (%) | | |
| | | | pass@1 | pass@5 | pass@10 | pass@1 | pass@5 | pass@10 |
|---|---|---|---|---|---|---|---|---|
| Foundational Models | Llama-3.1 | 8B | 40.7 | 60.6 | 65.5 | 19.3 | 34.7 | 37.9 |
| | Llama-3.1 | 405B | 56.5 | 64.4 | 72.4 | 38.9 | 45.8 | 51.7 |
| | Nemotron-4 | 340B | 41.7 | 47.2 | 48.3 | 18.9 | 20.7 | 20.7 |
| | GPT-3.5-turbo | - | 50.3 | 61.2 | 65.5 | 28.3 | 36.9 | 41.4 |
| | GPT-4o | - | 50.3 | 59.9 | 62.1 | 33.8 | 44.4 | 48.3 |
| Code Models | CodeLlama | 7B | 46.6 | 62.6 | 68.9 | 17.9 | 29.9 | 34.5 |
| | CodeQwen | 7B | 45.8 | 65.8 | 72.4 | 24.1 | 34.0 | 37.9 |
| | Starcoder2 | 15B | 38.3 | 81.0 | 94.7 | 15.5 | 37.6 | 45.7 |
| | DeepSeek-Coder | 6.7B | 51.4 | 64.4 | 68.9 | 23.1 | 29.3 | 34.5 |
| | DeepSeek-Coder-V2 | 16B | 51.4 | 57.8 | 58.6 | 33.1 | 37.1 | 37.9 |
| | DeepSeek-Coder-V2 | 236B | 63.4 | 78.1 | 79.3 | 34.5 | 50.2 | 55.1 |
| RTLCoder (Liu et al., 2023c) | Mistral | 7B | 64.6 | 73.7 | 78.3 | 24.5 | 37.3 | 42.3 |
| | DeepSeek-Coder | 6.7B | 73.4 | 83.9 | 86.2 | 35.8 | 40.3 | 43.1 |
| CodeV (Zhao et al., 2024) | CodeLlama | 7B | 79.0 | 89.2 | 89.9 | 39.4 | 50.3 | 53.1 |
| | DeepSeek-Coder | 6.7B | 78.3 | 87.4 | 89.1 | 42.4 | 51.5 | 53.2 |
| | CodeQwen | 7B | 78.8 | 89.5 | 92.4 | 36.6 | 53.3 | 61.3 |
| OriGen (Cui et al., 2024) | DeepSeek-Coder | 6.7B | - | - | - | - | 65.5 | - |
| Ours SDG-CC-Repair | CodeLlama | 7B | **85.7** | **93.9** | 94.8 | 42.6 | 52.9 | 58.2 |
| | DeepSeek-Coder | 6.7B | 84.3 | 92.9 | 95.4 | **53.1** | 58.8 | 62.6 |
| | Starcoder2 | 15B | 79.8 | **93.9** | **96.2** | 49.0 | **65.8** | **74.5** |

**RTLLM** (Lu et al., 2024) is an open-source benchmark designed for generating Register Transfer Level (RTL) code from natural language instructions. It evaluates models on syntax correctness, functional correctness, and design quality, offering a thorough analysis of model outputs.

## 4.3 RESULTS

**Main Results** Table 4 and Table 5 compare our models with baselines on VerilogEval and RTLLM. We mainly source baseline results from Zhao et al. (2024). For RTLLM we found a large variance with biased pass@5, thus we re-evalaute all models and report unbiased pass@k metric. We further rigorously evaluate the latest foundational and frontier code models, including Llama-3.1 (Dubey et al., 2024), DeepSeek-Coder-V2 (DeepSeek-AI et al., 2024), and GPT-4o. Recent foundational and frontier code models already reached competitive performance compared to previous efforts targeting Verilog code generation.

Compared to previous approaches like CodeV (Zhao et al., 2024), our models achieve comparable performance on VerilogEval-Machine and show significant improvements on benchmarks with human-like descriptions. Machine descriptions often provide detailed, line-by-line coding instructions, whereas human descriptions are high-level, integrating problem-solving skills and a deeper understanding of the hardware module's functionality. Enhancing the model's ability to handle human-like descriptions is crucial, as these more accurately reflect how designers interact with the models and set expectations for Verilog generation. Our fine-tuned Starcoder2-15B surpasses previous state-of-the-art results by 3.8%, 10.9%, and 6.6% in pass@1 metrics on VerilogEval-Machine, VerilogEval-Human, and RTLLM, respectively.

Table 6 highlights the effectiveness of our generated data fine-tuned on Starcoder2-15B. Our **CC** data enhances the model's ability to handle non-textual representations, leading to improved scores on VerilogEval-Human. Our targeted code **Repair** data boosts performance across all benchmarks, suggesting that the model has learned to generalize from code repair tasks and reduce similar errors during code completion.

Table 6: Ablation study on training data. Data quantity indicated in parentheses.

| Model | VerilogEval | | RTLLM v1.1 |
| | Machine | Human | Func |
| | pass@1 (%) | | pass@5 (%) |
|---|---|---|---|
| Starcoder2-15B | 68.7 | 37.7 | 37.6 |
| SDG (80.1k) | 75.2 | 54.7 | 62.1 |
| SDG-CC (108.6k) | 73.9 | 62.0 | 62.8 |
| SDG-CC-Repair (110.0k) | **81.9** | **68.0** | **65.8** |

**Improved Variability During Training** Figure 1b displays the pass rates for two consecutive checkpoints of Starcoder2-SDG-CC-Repair on VerilogEval-Human problems, sampled with a temperature of 0.8. Compared to Figure 1a, the updated model shows significant improvements by (1) moving previously unsolved problems into the solved category, including those with non-textual representations addressed by our correct-by-construction **CC** data, and (2) reducing the number of problems with large pass rate discrepancies, particularly where performance had degraded. The tar-

geted repair data has effectively mitigated the model's tendency to repeat common mistakes found in our **Repair** dataset, despite the noise inherent in synthetically generated **SDG** data.

**Scaling Data for Non-textual Representations** Figure 4 illustrates the scaling of correct-by-construction (**CC**) data and the fine-tuned Starcoder2-15B pass rate on problems involving non-textual representations. We expanded our testing to include strictly in-distribution test set, with each category containing around 50 problems. The results show that the model can quickly learn and comprehend these non-textual representations with as few as 4k training data samples, with the pass rate steadily improving as more data is provided. Additionally, the model demonstrates the ability to generalize to VerilogEval-NonText benchmark problems. While our models achieve near-perfect scores on KMap and FSM problems, they perform less effectively on Waveforms, suggesting that reverse engineering circuits from waveforms pose a greater challenge.

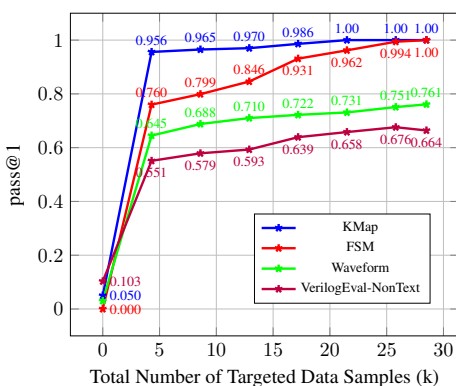

Figure 4: pass@1 on non-textual problems with total number of **CC** data with temperature 0.8.

**Ensuring Quality for Targeted Code Repair** The quality control mechanisms integrated into the data generation pipeline are crucial for improving model performance, particularly in correcting minor errors through targeted code repair. To evaluate the impact of these quality controls, we conducted an ablation study in Table 7, where we systematically removed each component of the targeted code repair generation pipeline and assessed the resulting model

Table 7: Ablation study on **Repair** data quality with Starcoder2-15B.

| Model | VerilogEval | | RTLLM v1.1 |
| --- | --- | --- | --- |
| | Machine | Human | Func |
| | pass@1 (%) | | pass@5 (%) |
| SDG-CC | 73.9 | 62.0 | 62.8 |
| SDG-CC-Repair | **81.9** | **68.0** | **65.8** |
| w/o self-consistency | 75.3 | 63.3 | 63.7 |
| w/o error report | 76.9 | 59.6 | 59.4 |

performance. Specifically, we eliminated the self-consistency checks that validate whether the generated error report effectively guides the LLMs in correcting mistakes. Additionally, we tested the removal of the error report entirely, substituting it with random errors injected into the open-source code by the LLMs. The benchmark results indicate a significant performance drop when these validation processes are excluded. These findings highlight the essential role of both the self-consistency checks and the targeted error report in improving the model's ability to correct errors.

## 5 RELATED WORK

**Synthetic Data Generation for Model Fine-tuning.** The performance of large language models (LLMs) hinge on the quality and diversity of their training data. To address the limitations of manual datasets, synthetic data generation methods (Wang et al., 2022; Xu et al., 2023) have been developed to automatically create instruction-following examples from LLMs, reducing reliance on human annotations. Various techniques enhance data quality: Wang et al. (2022) generates multiple reasoning traces and selects the most frequent output to improve robustness, while other approaches (Lightman et al., 2023; Zhang et al., 2024b) assess response quality based on these traces. Self-training methods utilize synthetic data for iterative fine-tuning, boosting reasoning capabilities (Singh et al., 2023; Feng et al., 2023). These advancements show how synthetic data can effectively scale and optimize models through iterative feedback.

**Large Language Models for Code Generation.** Recent breakthroughs in large language models (LLMs) have greatly enhanced their capability to tackle complex code generation tasks. Much of the research focuses on developing LLMs specialized for code by continuing their pretraining on code data (Guo et al., 2024; Bai et al., 2023; Roziere et al., 2023; DeepSeek-AI et al., 2024) from open-source repositories like GitHub (Kocetkov et al., 2022; Lozhkov et al., 2024) and commit histories (Muennighoff et al., 2023). Further improvements to these models come from reinforcement learning (Le et al., 2022) and more often instruction fine-tuning, which involves techniques

to address more complex coding problems (Luo et al., 2024b), increasing diversity with unlabeled open-source code (Wei et al., 2023; Yu et al., 2024; Wu et al., 2024), ensuring solution correctness through self-written tests (Chen et al., 2022), and validating and debugging code execution through interactions with LLM agents (Lei et al., 2024).

**Large Language Models for Verilog Coding.**   While most code LLMs target software languages, there is increasing interest in models for hardware description languages like Verilog, essential for chip design and verification (Liu et al., 2024). Previous work has addressed the challenge of limited data through various methods, including synthetic data generation (Liu et al., 2023c), multi-level summarization of open-source Verilog code (Zhao et al., 2024), and enhanced code augmentation with self-reflection based on compiler feedback (Tsai et al., 2023; Cui et al., 2024). Other approaches focus on improving functional correctness and circuit performance through Monte Carlo Tree Search (DeLorenzo et al., 2024) and discriminator-guided sampling (Pei et al., 2024).

## 6   DISCUSSIONS

In this work, we refer to *synthetic data generation* as methods of using large language models (LLMs) in data generation. While our approach—ensuring correctness through correct-by-construction—could also be considered "synthetic" and resembles methods explored in works like AlphaGeometry (Trinh et al., 2024), our problems are much simpler and on a smaller scale. Our observations about the variability of models on specific problems align with the findings of  Meta AI (2024), where "*the model knows how to produce the right answer, but it does not know how to select it*." Instead of striving for absolute data correctness, preference learning (Rafailov et al., 2024; Ethayarajh et al., 2024) or reinforcement learning (Bai et al., 2022; Le et al., 2022), we generate targeted repair data by analyzing errors and re-create such scenarios by injecting similar errors into open-source code, somewhat analogous to how humans consolidate memories during sleep by integrating new information with past experiences (Walker & Stickgold, 2004; Stickgold, 2005). Further discussions on the generalizability and broader impact of our work are provided in Appendix B.

## 7   CONCLUSION

This paper addresses key challenges in Verilog code generation with correct-by-construction data generation and targeted code repair data strategies. We identified significant issues with synthetic data generation, including difficulties with non-textual representations and variability in performance during training across benchmarks. To address these challenges, we generated data that is correct-by-construction and create targeted repair data by injecting errors to open-source code. Our approach led to substantial improvements, with models fine-tuned using our methods achieving state-of-the-art results on VerilogEval and RTLLM benchmarks. These advancements highlight the effectiveness of our strategies in enhancing model performance in Verilog code generation.

**Reproducibility Statement**   We provide the following details: evaluation benchmarks in Appendix A.3, examples of the process for generating targeted code repair data in Appendix C, and data examples from correct-by-construction targeting non-textual representations in Appendix D. Additionally, we include prompt templates used for data generation in Appendix E. Our data generation pipeline is available: `https://github.com/NVlabs/CraftRTL`.

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

# A    DETAILED RESULTS

## A.1    OUR MODELS

We present our models' results on Verilog benchmarks tested with temperatures 0.2 and 0.8. We ablate across different data blends, with **SDG** indicating using LLM synthetic generated data in Section 2.1, **CC** indicating correct-by-construction data targeting non-textual representations in Section 3.1, and **Repair** representing our targeted code repair dataset in Section 3.2.

Our results for RTLLM use the open-source Icarus Verilog simulator[1] to check syntax and functional pass rates. This might lead to lower pass rate scores compared to previous work that used Synopsys VCS, as Icarus Verilog does not support all syntax.

Table 8: Results for our models, across different dataset and temperature on VerilogEval.

| Model | Dataset | Temperature | VerilogEval (Liu et al., 2023b) | | | | | |
|---|---|---|---|---|---|---|---|---|
| | | | Machine (%) | | | Human (%) | | |
| | | | pass@1 | pass@5 | pass@10 | pass@1 | pass@5 | pass@10 |
| Starcoder2-15b | SDG | 0.2 | 75.2 | 79.2 | 80.1 | 54.7 | 60.1 | 61.2 |
| | | 0.8 | 73.7 | 84.0 | 86.1 | 47.4 | 61.9 | 64.8 |
| | SDG-CC | 0.2 | 73.9 | 78.1 | 79.5 | 62.0 | 65.6 | 67.0 |
| | | 0.8 | 72.9 | 84.1 | 87.1 | 58.5 | 70.3 | 73.7 |
| | SDG-CC-Repair | 0.2 | 81.9 | 84.2 | 85.0 | 68.0 | 71.7 | 72.0 |
| | | 0.8 | 78.1 | 86.9 | 88.1 | 64.1 | 72.4 | 74.6 |
| DeepSeek-6.7b-Instruct | SDG | 0.2 | 73.4 | 77.8 | 78.9 | 48.3 | 53.2 | 54.5 |
| | | 0.8 | 71.4 | 82.5 | 85.4 | 44.0 | 58.1 | 62.3 |
| | SDG-CC | 0.2 | 72.6 | 78.2 | 79.3 | 58.5 | 62.6 | 63.5 |
| | | 0.8 | 70.2 | 83.1 | 85.4 | 56.3 | 67.0 | 70.7 |
| | SDG-CC-Repair | 0.2 | 77.8 | 82.7 | 83.4 | 65.4 | 67.7 | 68.2 |
| | | 0.8 | 75.2 | 85.5 | 88.1 | 61.6 | 70.0 | 72.1 |
| CodeLlama-7b-Instruct | SDG | 0.2 | 74.5 | 77.9 | 78.8 | 45.3 | 50.3 | 51.5 |
| | | 0.8 | 71.2 | 82.6 | 85.1 | 42.6 | 55.6 | 59.0 |
| | SDG-CC | 0.2 | 74.2 | 77.4 | 78.1 | 55.1 | 61.0 | 62.4 |
| | | 0.8 | 70.0 | 81.2 | 83.7 | 51.6 | 64.4 | 67.7 |
| | SDG-CC-Repair | 0.2 | 78.1 | 81.5 | 81.7 | 63.1 | 66.2 | 66.8 |
| | | 0.8 | 73.7 | 85.5 | 87.8 | 58.1 | 67.8 | 69.7 |

Table 9: Results for our models, across different dataset and temperature on RTLLM.

| Model | Dataset | Temperature | RTLLM v1.1 Lu et al. (2024) | | | | | |
|---|---|---|---|---|---|---|---|---|
| | | | Syntax (%) | | | Func. (%) | | |
| | | | pass@1 | pass@5 | pass@10 | pass@1 | pass@5 | pass@10 |
| Starcoder2-15b | SDG | 0.2 | 78.1 | 86.5 | 90.1 | 49.0 | 60.4 | 66.3 |
| | | 0.8 | 77.1 | 89.0 | 94.1 | 43.8 | 62.1 | 68.0 |
| | SDG-CC | 0.2 | 78.3 | 89.3 | 92.7 | 45.5 | 58.3 | 62.0 |
| | | 0.8 | 76.9 | 92.6 | 95.5 | 38.4 | 62.8 | 70.4 |
| | SDG-CC-Repair | 0.2 | 79.8 | 87.9 | 90.5 | 49.0 | 59.1 | 62.6 |
| | | 0.8 | 79.3 | 93.9 | 96.2 | 45.3 | 65.8 | 74.5 |
| DeepSeek-6.7b-Instruct | SDG | 0.2 | 79.3 | 86.8 | 90.5 | 40.3 | 45.9 | 49.6 |
| | | 0.8 | 76.6 | 92.5 | 96.2 | 40.0 | 53.8 | 63.6 |
| | SDG-CC | 0.2 | 73.6 | 84.5 | 86.0 | 44.3 | 52.2 | 54.3 |
| | | 0.8 | 76.7 | 90.5 | 93.8 | 39.5 | 56.4 | 63.1 |
| | SDG-CC-Repair | 0.2 | 84.3 | 92.2 | 93.0 | 53.1 | 58.8 | 60.3 |
| | | 0.8 | 80.0 | 92.9 | 95.4 | 45.5 | 57.9 | 62.6 |
| CodeLlama-7b-Instruct | SDG | 0.2 | 74.0 | 82.5 | 86.8 | 30.0 | 33.9 | 35.8 |
| | | 0.8 | 70.9 | 89.1 | 94.5 | 34.0 | 47.2 | 52.8 |
| | SDG-CC | 0.2 | 75.0 | 90.2 | 94.6 | 39.7 | 44.4 | 47.2 |
| | | 0.8 | 76.4 | 93.9 | 96.3 | 35.5 | 47.6 | 52.7 |
| | SDG-CC-Repair | 0.2 | 85.7 | 93.9 | 94.8 | 42.6 | 49.4 | 51.2 |
| | | 0.8 | 80.3 | 93.9 | 94.8 | 36.9 | 52.9 | 58.2 |

---

[1]https://github.com/steveicarus/iverilog

## A.2 FOUNDATIONAL AND FRONTIER CODE MODELS

We present detailed results on recent foundational and frontier code models. We also re-evaluate all models on RTLLM using unbiased pass@k metric.

Table 10: Results on foundational and code models on VerilogEval.

| Type | Model | Size | Temp | VerilogEval (Liu et al., 2023b) | | | | | |
|---|---|---|---|---|---|---|---|---|---|
| | | | | Machine (%) | | | Human (%) | | |
| | | | | pass@1 | pass@5 | pass@10 | pass@1 | pass@5 | pass@10 |
| Foundational Models | Llama-3.1 | 8B | 0.2 | 48.7 | 66.2 | 70.6 | 26.9 | 36.9 | 40.4 |
| | | | 0.8 | 42.1 | 67.3 | 74.1 | 23.0 | 37.8 | 44.2 |
| | Llama-3.1 | 70B | 0.2 | 66.7 | 73.8 | 76.9 | 48.7 | 53.6 | 55.1 |
| | | | 0.8 | 64.5 | 77.7 | 80.4 | 48.0 | 57.0 | 60.9 |
| | Llama-3.1 | 405B | 0.2 | 67.3 | 72.8 | 74.1 | 51.9 | 57.0 | 58.9 |
| | | | 0.8 | 66.4 | 75.1 | 76.9 | 53.8 | 61.0 | 62.8 |
| | Nemotron-4 | 340B | 0.2 | 53.0 | 59.1 | 61.5 | 43.1 | 43.9 | 44.9 |
| | | | 0.8 | 50.8 | 60.3 | 62.2 | 40.8 | 48.3 | 50.0 |
| | GPT-3.5-turbo | - | 0.2 | 58.0 | 66.4 | 68.5 | 31.2 | 39.4 | 41.7 |
| | | | 0.8 | 56.6 | 74.0 | 77.6 | 28.9 | 44.1 | 47.4 |
| | GPT-4 | - | 0.2 | 53.2 | 63.7 | 66.4 | 36.1 | 43.5 | 46.2 |
| | | | 0.8 | 35.3 | 53.4 | 58.9 | 35.2 | 53.4 | 58.9 |
| | GPT-4-turbo | - | 0.2 | 57.8 | 66.7 | 70.6 | 54.1 | 61.2 | 62.8 |
| | | | 0.8 | 56.9 | 69.5 | 73.4 | 53.6 | 63.6 | 66.7 |
| | GPT-4o | - | 0.2 | 65.9 | 68.9 | 69.2 | 57.1 | 61.3 | 62.2 |
| | | | 0.8 | 62.9 | 71.4 | 72.7 | 55.4 | 63.9 | 66.7 |
| Code Models | Starcoder2 | 15B | 0.2 | 68.7 | 76.7 | 78.6 | 37.7 | 48.3 | 51.1 |
| | | | 0.8 | 57.7 | 82.3 | 88.5 | 29.1 | 50.6 | 57.2 |
| | DeepSeek-Coder-V2 | 16B | 0.2 | 67.4 | 74.6 | 76.2 | 46.9 | 53.3 | 54.5 |
| | | | 0.8 | 65.6 | 78.3 | 81.8 | 46.3 | 55.9 | 58.9 |
| | DeepSeek-Coder-V2 | 236B | 0.2 | 68.2 | 72.7 | 75.0 | 56.4 | 60.7 | 64.3 |
| | | | 0.8 | 66.5 | 74.1 | 76.2 | 54.8 | 62.2 | 66.0 |

Table 11: Results on foundational and code models on RTLLM.

| Type | Model | Size | Temp | RTLLM v1.1 (Lu et al., 2024) | | | | | |
|---|---|---|---|---|---|---|---|---|---|
| | | | | Syntax (%) | | | Func. (%) | | |
| | | | | pass@1 | pass@5 | pass@10 | pass@1 | pass@5 | pass@10 |
| Foundational Models | Llama-3.1 | 8B | 0.2 | 39.7 | 53.1 | 55.2 | 19.3 | 25.8 | 27.6 |
| | | | 0.8 | 40.7 | 60.6 | 65.5 | 17.6 | 34.7 | 37.9 |
| | Llama-3.1 | 70B | 0.2 | 47.9 | 51.7 | 55.2 | 34.1 | 34.5 | 34.5 |
| | | | 0.8 | 48.9 | 57.6 | 58.6 | 29.6 | 31.0 | 31.0 |
| | Llama-3.1 | 405B | 0.2 | 56.5 | 63.9 | 65.5 | 38.9 | 45.0 | 48.3 |
| | | | 0.8 | 52.1 | 64.4 | 72.4 | 35.8 | 45.8 | 51.7 |
| | Nemotron-4 | 340B | 0.2 | 41.7 | 47.2 | 48.3 | 14.1 | 15.5 | 17.2 |
| | | | 0.8 | 41.7 | 46.3 | 48.3 | 18.9 | 20.7 | 20.7 |
| | GPT-3.5-turbo | - | 0.2 | 50.3 | 58.2 | 58.6 | 28.3 | 36.9 | 41.4 |
| | | | 0.8 | 48.2 | 61.2 | 65.5 | 24.1 | 36.9 | 41.4 |
| | GPT-4 | - | 0.2 | 49.3 | 65.9 | 68.9 | 30.0 | 44.4 | 48.3 |
| | | | 0.8 | 42.8 | 61.2 | 65.5 | 25.9 | 40.0 | 44.8 |
| | GPT-4-turbo | - | 0.2 | 38.9 | 44.8 | 48.3 | 27.2 | 35.1 | 37.9 |
| | | | 0.8 | 40.3 | 48.8 | 51.7 | 27.5 | 40.2 | 44.8 |
| | GPT-4o | - | 0.2 | 50.3 | 59.9 | 62.1 | 33.8 | 44.4 | 48.3 |
| | | | 0.8 | 47.5 | 63.2 | 66.7 | 31.3 | 44.1 | 48.3 |
| Code Models | CodeLlama | 7B | 0.2 | 46.6 | 62.6 | 68.9 | 17.9 | 29.9 | 34.5 |
| | | | 0.8 | 34.8 | 59.7 | 68.9 | 13.4 | 25.9 | 31.0 |
| | CodeQwen | 7B | 0.2 | 45.8 | 55.8 | 58.6 | 24.1 | 33.1 | 37.9 |
| | | | 0.8 | 45.5 | 65.7 | 72.4 | 22.4 | 34.0 | 37.9 |
| | Starcoder2 | 15B | 0.2 | 38.3 | 77.5 | 86.3 | 15.5 | 37.6 | 44.6 |
| | | | 0.8 | 31.6 | 81.0 | 94.7 | 11.0 | 34.2 | 45.7 |
| | DeepSeek-Coder | 6.7B | 0.2 | 51.4 | 62.6 | 65.5 | 23.1 | 26.8 | 27.6 |
| | | | 0.8 | 49.7 | 64.4 | 68.9 | 21.0 | 29.3 | 34.5 |
| | DeepSeek-Coder-V2 | 16B | 0.2 | 51.4 | 51.7 | 51.7 | 33.1 | 34.5 | 34.5 |
| | | | 0.8 | 51.4 | 57.8 | 58.6 | 30.0 | 37.1 | 37.9 |
| | DeepSeek-Coder-V2 | 236B | 0.2 | 63.4 | 73.0 | 79.3 | 34.5 | 44.9 | 52.9 |
| | | | 0.8 | 61.8 | 78.1 | 79.3 | 32.9 | 50.2 | 55.1 |

## A.3 Details on Evaluations

We format the prompt input as follows for VerilogEval, where the *detail_description* is the problem description (Machine or Human) and *prompt* field is the problem module header. We include module headers to avoid confusion on the signals naming.

```
prompt = f"{task['detail_description'].strip()}\n\n{task['prompt'].strip()}"
```

An example of *mux2to1* in VerilogEval-Human:

```
Create a 2—1 multiplexer. When sel=0, choose a. When sel=1, choose b.

module top_module (
        input a,
        input b,
        input sel,
        output out
);
```

We use similar templates for RTLLM v1.1, where we extract the top module header from the reference solution and provide it as input. Below is an example of *adder_8bit*:

```
Please act as a professional verilog designer.

Implement a module of an 8—bit adder with multiple bit—level adders in combinational
    logic.

Module name:
    adder_8bit
Input ports:
    a[7:0]: 8—bit input operand A.
    b[7:0]: 8—bit input operand B.
    cin: Carry—in input.
Output ports:
    sum[7:0]: 8—bit output representing the sum of A and B.
    cout: Carry—out output.

Implementation:
The module utilizes a series of bit—level adders (full adders) to perform the addition
     operation.

Give me the complete code.

module adder_8bit(
    input [7:0] a, b,
    input cin,
    output [7:0] sum,
    output cout);
```

We use default chat templates and default system prompts for open-source models tested. For GPT models from OpenAI, we use the following system prompt:

```
Please act as a professional verilog designer.
```

We post-process model responses to extract code. We extract content enclosed by triple backticks and remove the language identifier (Verilog). We then extract code enclosed in `module` and `endmodule` keywords with `response.find('module')` and `response.rfind('endmodule')`. If the extracted code does not include a module header, the reference solution's module header will be prepended. The code is then tested with the provided testbenches with the Icarus Verilog (iverilog) simulator to evaluate for syntax and functional correctness. This might lead to lower pass rate scores for RTLLM compared to previous work that used Synopsys VCS, as Icarus Verilog does not support all syntax.

## A.4 VERILOGEVAL-NONTEXT

We select the following 45 problems from VerilogEval-Human that consists of non-textual representations in their problem descriptions:

*2012_q1g, 2012_q2b, 2012_q2fsm, 2013_q2afsm, 2014_q3bfsm, 2014_q3c, always_nolatches, circuit1, circuit10, circuit2, circuit3, circuit4, circuit5, circuit6, circuit7, circuit8, circuit9, ece241_2013_q7, ece241_2014_q3, ece241_2014_q5b, fsm1, fsm1s, fsm2, fsm2s, fsm3, fsm3comb, fsm3onehot, fsm3s, fsm_onehot, fsm_ps2data, kmap1, kmap2, kmap3, kmap4, m2014_q3, m2014_q6, m2014_q6b, m2014_q6c, mt2015_q4, mt2015_q4a, mt2015_q4b, review2015_fsmonehot, rule110, rule90, truthtable1*

## A.5 TEMPLATE PROBLEMS FOR CORRECT-BY-CONSTRUCTION DATA

When generating correct-by-construction **CC** data, we select 11 problems from VerilogEval-NonText to use as representative templates for constructing our prompts. To prevent contamination, we ensure that benchmark problems are excluded from our data. While our prompts will resemble those of the selected problems, the non-textual representations and solutions will differ. Additionally, to prevent overfitting to specific prompt templates, we use LLMs to rewrite the problem instructions for 20% of our data. Furthermore, we create validation test problems that are strictly in-distribution, based on the chosen problems.

**Karnaugh Maps and Truth Tables:** *kmap1, m2014_q3, truthtable1*.

**State Transition Graphs and Tables:** *2012_q2b, 2014_q3c, ece241_2014_q5b, fsm3comb, fsm3onehot, fsm_onehot, m2014_q6b, m2014_q6c*.

**Waveforms:** We do not base our data on any benchmark problems specifically.

## A.6 SCALING REPAIR DATA

As shown in Table 12, a carefully filtered dataset of 1.4k samples achieves comparable performance to a 7.8k dataset. This suggests that merely increasing the dataset size by injecting the same types of errors does not contribute meaningfully to improving model performance.

Table 12: Scaling **Repair** data.

| Model | VerilogEval | | RTLLM v1.1 |
|---|---|---|---|
| | Machine | Human | Func |
| | pass@1 (%) | | pass@5 (%) |
| SDG-CC | 73.9 | 62.0 | 62.8 |
| SDG-CC-Repair 1k | 81.9 | **68.0** | **65.8** |
| SDG-CC-Repair 7k | **82.2** | 67.4 | 64.5 |

## A.7 ITERATIVE CODE REPAIR

We conduct a second iteration by generating 2.7k repair data for the model based on the **Repair** data from the first iteration. As shown in Table 13, performance mostly saturates after this initial iteration. We suspect that the remaining issues are likely due to significant errors that are challenging to correct.

Table 13: Iterative code repair.

| Model | VerilogEval | | RTLLM v1.1 |
|---|---|---|---|
| | Machine | Human | Func |
| | pass@1 (%) | | pass@5 (%) |
| SDG-CC | 73.9 | 62.0 | 62.8 |
| SDG-CC-Repair Iter 1 | **81.9** | 68.0 | **65.8** |
| SDG-CC-Repair Iter 2 | 81.3 | **68.1** | 65.6 |

## A.8 DIVERSITY OF GENERATED CODE

We assess the diversity of the code generated by our models. We measure this diversity using BLEU score, Jaccard similarity, and abstract tree edit distance (TSED) in Song et al. (2024). The VerilogEval-Human problems are categorized into NonText and Text, as described in Appendix A.4. For each problem, we compute the average code diversity score across sampled codes for the same problem and report the mean score for all problems. For TSED, we use PyVerilog (Takamaeda-Yamazaki, 2015) to extract the abstract syntax tree, and codes that fail syntax checks are excluded from the analysis.

Table 14 presents the results on code diversity. We sample 20 solutions with temperature of 0.8 for each model. We observe that fine-tuned models generally show a decrease in code diversity for both Text and NonText problems. This reduction is expected, as BLEU and Jaccard metrics account for both correct and incorrect code solutions, and there are often multiple ways to implement a correct

solution. When comparing our fine-tuned models with GPT-4o, code diversity is similar for Text problems, but our models exhibit poor diversity for NonText problems. This is anticipated, given that the **CC** training dataset for NonText problems is generated using correct-by-construction methods and follows similar templates for Verilog code. However, our models demonstrate comparable diversity to GPT-4o for Text problems, particularly in TSED metric.

Table 14: Diversity of generated code solutions on VerilogEval-Human sampled with temperature of 0.8. Lower scores indicate higher diversity.

| Type | Models | Text | | | NonText | | |
|---|---|---|---|---|---|---|---|
| | | Jaccard | BLEU | TSED | Jaccard | BLEU | TSED |
| Pretrained Models | CodeLlama | 0.5330 | 0.3808 | 0.4255 | 0.4707 | 0.2507 | 0.3521 |
| | DeepSeek-Coder | 0.6606 | 0.5454 | 0.5956 | 0.6548 | 0.3797 | 0.3847 |
| | Starcoder2 | 0.7724 | 0.5084 | 0.5520 | 0.7212 | 0.3607 | 0.4020 |
| | GPT-4o | 0.6798 | 0.6633 | 0.6906 | 0.7390 | 0.6376 | 0.6137 |
| Ours SDG-CC-Repair | CodeLlama | 0.6848 | 0.5992 | 0.6354 | 0.8583 | 0.7242 | 0.7158 |
| | DeepSeek-Coder | 0.6828 | 0.6040 | 0.6319 | 0.8308 | 0.6866 | 0.6598 |
| | Starcoder2 | 0.7018 | 0.6381 | 0.6721 | 0.8799 | 0.7750 | 0.7740 |

| Type | Models | VerilogEval-Human (Overall) | | |
|---|---|---|---|---|
| | | Jaccard | BLEU | TSED |
| Pretrained Models | CodeLlama | 0.5155 | 0.3441 | 0.4156 |
| | DeepSeek-Coder | 0.6590 | 0.4987 | 0.5505 |
| | Starcoder2 | 0.7580 | 0.4667 | 0.5198 |
| | GPT-4o | 0.6965 | 0.6561 | 0.6802 |
| Ours SDG-CC-Repair | CodeLlama | 0.7333 | 0.6345 | 0.6515 |
| | DeepSeek-Coder | 0.7246 | 0.6273 | 0.6379 |
| | Starcoder2 | 0.7512 | 0.6767 | 0.6942 |

## A.9    ERROR TYPES OF LLM GENERATED ERROR REPORTS

Table 15: Error types of LLM generated error reports.

| Error Type | #Errors | One-line Description |
|---|---|---|
| Vector Concatenation | 15.3% | Errors during vector concatenation or bit slicing. |
| Incorrect Initialization | 13.1% | Missing or faulty initialization of registers or signals. |
| Boolean Logic Flaws | 12.4% | Logical inconsistencies or errors in combinational logic expressions. |
| Shift Operation Faults | 10.2% | Misaligned or unintended behavior during shift operations. |
| Timing Violations | 10.2% | Errors where signal propagation violates timing requirements. |
| KMap Misinterpretation | 8.8% | Incorrect derivation of Boolean expressions from Karnaugh maps. |
| Latch Hazards | 6.5% | Unintended latches caused by missing or faulty conditions. |
| Bit Manipulation Bugs | 7.3% | Errors in operations like masking, flipping, or extracting specific bits. |
| Casez Priority Conflicts | 4.4% | Ambiguities or conflicts in casez or case statements. |
| Nested Loop Design Flaws | 3.7% | Incorrect or inefficient nested loop designs. |
| Others | 8.1% | Miscellaneous errors not covered above. |

Table 15 shows the distribution of common error types in LLM-generated error reports, along with brief one-line descriptions. Most of these "minor" errors occur in solvable problems and stem from hardware-specific concepts (e.g., shift operations, timing violations) and Verilog related issues un-

common in software languages (e.g., latch hazards, casez priority conflicts). When generating targeted repair training data, we randomly sample detailed error reports and open-source code snippets, ensuring the error type distribution in training aligns with their natural occurrences.

## A.10 Details on Figure 1

In Section 2.3 we discussed our findings on training variability in learning outcomes for specific benchmark problems. To analyze this, we saved checkpoints every 64 gradient steps during training and tracked the pass rates of specific benchmarks. Our training process is limited to a single epoch, as fur-

Table 16: Checkpoints of Figure 1.

| Model | checkpoint1 | | checkpoint2 | |
|---|---|---|---|---|
| | Steps | Epoch | Steps | Epoch |
| SDG | 256 | 0.82 | 320 | 1.0 |
| SDG-CC-Repair | 386 | 0.86 | 448 | 1.0 |

ther training was found to be not helpful. We classify problems with pass rates exceeding 67% as solvable, and those below 33% as unsolvable. For the visualizations in Figure 1 we selected the final two saved checkpoints, detailed in Table 16. The ideal outcome is not merely reduced variability but also less degradations and improved accuracy: specifically, most problems in checkpoint2 should show higher pass rates than checkpoint1, assuming that training on additional data enhances model performance. However, as shown in Figure 1a training on SDG data results in a significant degradation of pass rates for many problems between checkpoint1 and checkpoint2. In contrast Figure 1b demonstrates reduced degradation and improvement in more problems. We further elaborate such findings in Table 17, where we display pass rates for selected benchmark problems with high volatility from VerilogEval-Human throughout the training progression.

Table 17: We displays pass rates for selected benchmark problems from VerilogEval-Human throughout the training progression. Each entry shows the pass rate for SDG-CC-Repair (SDG), with SDG results in parentheses.

| Problem | Step 64 | Step 128 | Step 256 | Step 320 | Step 386 | Step 448 |
|---|---|---|---|---|---|---|
| m2014_q4h | 1.0 (1.0) | 1.0 (0.9) | 1.0 (0.967) | 1.0 (0.875) | 1.0 (-) | 1.0 (-) |
| always_nolatches | 1.0 (0.867) | 1.0 (0.9) | 1.0 (0.6) | 1.0 (0.833) | 1.0 (-) | 1.0 (-) |
| vectorr | 1.0 (0.633) | 1.0 (0.925) | 1.0 (0.467) | 0.95 (0.925) | 1.0 (-) | 1.0 (-) |
| fsm2s | 1.0 (0.8) | 1.0 (0.8334) | 0.8 (0.775) | 1.0 (0.967) | 1.0 (-) | 1.0 (-) |
| fsm3comb | 1.0 (0.0) | 0.95 (1.0) | 0.5 (0.533) | 1.0 (0.233) | 1.0 (-) | 1.0 (-) |

We believe such volatility primarily is due to noise in SDG data where we can not verify solution correctness. Because of the difficulties of verifying coding solutions in hardware descriptive languages, we instead generate targeted repair data for LLMs to learn to mitigate common errors which have shown to generalize to writing correct code during completion. To the best of our knowledge, we are the first work to describe such findings and provide an effective solution.

## B  Further Discussions and Broader Impacts

In this section, we provide further discussions to address concerns regarding the novelty, generalizability, and significance of our proposed methods. We offer clarifications to highlight the relevance and broader impact of our work, underscoring its value to the broad research community.

### B.1  Generalizability of Correct-by-Construction Data Generation

Our approach to curating correct-by-construction data is largely inspired by Trinh et al. (2024), who introduced a mathematically rigorous method utilizing symbolic deduction engines to construct synthetic training data, significantly improving LLM capabilities in solving Olympiad geometry problems. Similarly, our method ensures the correctness of problems and solutions through a custom-designed data generation pipeline, leveraging custom-designed solvers to generate accurate solutions to their corresponding problems. In contrast to methods distilling LLM responses like Self-Instruct (Wang et al., 2022), our correct-by-construction approach ensures data quality and solution accuracy without relying on strong LLM performance on downstream tasks. We hope that our mathematically rigorous approach to generating synthetic data can further inspire future work

on improving LLMs general capabilities in areas such as math, coding, and symbolic reasoning. Moreover, we recognize that adapting these methods to other domains may require human tuning to identify the best data generation method, and we note that automating this process for scalability could be a promising future research direction.

## B.2 Novelty and Generalizability of Targeted Code Repair

Our analysis show that LLMs frequently make "minor" errors in Verilog coding, often correctable within few lines of code. We attribute this primarily to the LLMs' insufficient training in comprehending problem descriptions and instructions alongside their correct solutions. Prior research has tackled this challenge by improving data quality. For instance, Chen et al. (2022) filters incorrect code using tests generated by LLMs, while Zhang et al. (2024c) creates preference learning datasets by ranking code through self-validation. Lei et al. (2024) focus on generating fine-tuning data through code completion, test validation, and debugging with LLM agents, while Le et al. (2022) trained reward models based on compilation and unit test outcomes to enhance LLM performance via reinforcement learning. However, low-resource languages face additional obstacles due to limited data availability, making it particularly difficult to synthesize unit tests directly in these languages. To address this issue, Cassano et al. (2024) introduced lightweight compilers to translate test cases from source to target languages.

Verilog coding encounters challenges typical of low-resource languages, compounded additional domain-specific challenges as a hardware description language rather than a conventional programming language. Its unique characteristics pose significant barriers to knowledge transfer from high-resource languages, as highlighted in studies on execution performance in parallel programming (Nichols et al., 2024) and high-performance computing extensions (TehraniJamsaz et al., 2024). To address these challenges, we propose a novel pipeline for generating targeted code repair data. While automatic code repair has been extensively studied, most existing methods focus on widely-used programming languages (Xia et al., 2023), relying on data of buggy code and fixes from open-source repositories (Tufano et al., 2019; Just et al., 2014). In contrast, our pipeline utilizes a small set of well-curated benchmarks and testbench to automate the generation of error reports, quality assurance, and augmentation of training datasets by injecting similar errors into open-source code. Our results highlight the effectiveness of this approach, which is language agnostic and can be adapted to other low-resource and domain-specific programming languages.

## B.3 Significance of Non-Textual Data Representations in Hardware Design

In this work, we emphasize the significance of non-textual data representations, specifically Karnaugh maps, state-transition diagrams, and waveforms, for accurately capturing hardware functionality. These representations are widely utilized by hardware designers to mitigate the ambiguity and verbosity inherent in natural language descriptions. While they may be specific to hardware design, they are not limited to Verilog and can be applied to various domain-specific languages (DSLs) for hardware design. This is supported by Batten et al. (2024), who leveraged similar non-textual representations from VerilogEval-Human to evaluate the performance of LLMs on several Python-embedded hardware design DSLs.

In this study, we focus exclusively on limited representations, which constitute a significant portion (45 problems, approximately 30%) of all problems in the VerilogEval-Human benchmark (details in Appendix A.4). We exclude other types of non-textual representations due to the lack of a suitable benchmark for evaluating LLMs in Verilog coding. Chang et al. (2024) emphasize the importance of non-textual representations, particularly visual representations, in describing hardware designs. While their work targets visual-language models and is therefore beyond the scope of this study, we recognize that similar methodologies of our work such as correct-by-construction methods could be employed to generate training data for visual representations, such as circuit schematics, data flow diagrams, and state transition graphs.

## C  EXAMPLES OF TARGETED CODE REPAIR DATA

---

**Problem description:** *vector3* from VerilogEval-Machine

The top module contains 6 ports that are fed into a concatenation which outputs 4 ports.
The first input "a" is fed into the first port in the concatenation, the second input "b" is fed into the second port of the concatenation, the third input "c" is fed into the third port of the concatenation, the fourth input "d" is fed into the fourth port of the concatenation, the fifth input "e" is fed into the fifth port of the concatenation, the sixth input "f" is fed into the sixth port of the concatenation. The last bit of the concatenation is 2'b11. This value is fed in to the last port of the concatenation.

```
module top_module (
  input [4:0] a,
  input [4:0] b,
  input [4:0] c,
  input [4:0] d,
  input [4:0] e,
  input [4:0] f,
  output [7:0] w,
  output [7:0] x,
  output [7:0] y,
  output [7:0] z
);
```

---

```
  assign {w, x, y, z} = {a, b, c, d, e, f
    , 2'b11};

endmodule
```

```
// Concatenate the inputs with the last
    bit as 2'b11
  assign {w, x, y, z} = {2'b11, a, b, c,
    d, e, f};

endmodule
```

Correct Code                    Error Code

---

**LLM generated Error Report**

Error Type: Incorrect vector concatenation and splitting

Category: Combinatorial: wiring

Description: Steps to repair the erroneous implementation:

1. Concatenating the two 1 bits at the beginning of the concatenated vector.
2. Assign the output vectors from the concatenated vector in the correct order and bit ranges

For example:

```
assign {w, x, y, z} = {2'b11, a, b, c, d, e, f};
```

To correct the implementation, you should concatenate the last bit as '2'b11' on the right, as shown in the correct implementation:

```
assign {w, x, y, z} = {a, b, c, d, e, f, 2'b11};
```

---

**LLM constructed Repair data from open-source code**

**Input:**

#### 1.  Problem Description
You are given a Verilog module that demonstrates the use of bit slicing and concatenation in a loop. The goal is to correctly display specific portions of a 32-bit register using a for-loop and explicit slicing. However, there is a commonly made error in the concatenation and bit slicing order that causes incorrect results to be displayed. Your task is to identify and fix the error in the code.

---

#### 2. Erroneous Implementation

```verilog
module block;
    reg [31:0]  data;
    int         i;

    initial begin
        data = 32'hFACE_CAFE;

        // Loop to display 8-bit segments of the data register
        for (i = 0; i < 4; i++) begin
            // Incorrectly reversing the bit order in the concatenation
            $display ("data[8*%0d +: 8] = 0x%0h", i, data[8*(3-i) +: 8]);
        end

        // Incorrect slicing order, placing higher bits in lower indexes
        $display ("data[7:0]   = 0x%0h", data[31:24]);
        $display ("data[15:8]  = 0x%0h", data[23:16]);
        $display ("data[23:16] = 0x%0h", data[15:8]);
        $display ("data[31:24] = 0x%0h", data[7:0]);
    end
endmodule
```

In this erroneous implementation:
- The loop erroneously reverses the order of the 8-bit slices.
- The explicit bit-slicing outside of the loop has the slices ordered incorrectly, assigning the MSB to the least significant part and vice versa.

#### 3. Hints for Fixing
1. **Correct the loop:**
The loop currently reverses the order of the 8-bit slices. Instead of accessing 'data[8*(3-i) +: 8]', you should access 'data[8*i +: 8]' to display the correct slices of the register.
2. **Fix the slicing outside of the loop:**
Ensure that the LSB ('data[7:0]') corresponds to the lower bits of the 'data' register, and the MSB ('data[31:24]') corresponds to the higher bits. Correct the slicing order for accurate display.

The corrected implementation should ensure that both the loop and explicit bit-slicing access the register correctly, without reversing the bit order.

**Output:**

```verilog
module block;
    reg [31:0]  data;
    int         i;
    initial begin
        data = 32'hFACE_CAFE;
        for (i = 0; i < 4; i++) begin
            $display ("data[8*%0d +: 8] = 0x%0h", i, data[8*i +: 8]);
        end
        $display ("data[7:0]   = 0x%0h", data[7:0]);
        $display ("data[15:8]  = 0x%0h", data[15:8]);
        $display ("data[23:16] = 0x%0h", data[23:16]);
        $display ("data[31:24] = 0x%0h", data[31:24]);
    end
endmodule
```

Figure 5: An example demonstrating the process for targeted code repair. In this example, training checkpoints during training would have the model sometimes generated correct and error solutions. We use LLMs to first summarize the errors for a detailed Error Report and then inject the errors to open-source code to construct Repair data.

---

**Problem description:** *shift4* from VerilogEval-Human

Build a 4-bit shift register (right shift), with asynchronous positive edge triggered areset, synchronous active high signals load, and enable.
// (1) areset: Resets shift register to zero.
// (2) load: Loads shift register with data[3:0] instead of shifting.
// (3) ena: Shift right (q[3] becomes zero, q[0] is shifted out and disappears).
// (4) q: The contents of the shift register. If both the load and ena inputs are asserted (1), the load input has higher priority.

```verilog
module top_module(
  input clk,
  input areset,
  input load,
  input ena,
  input [3:0] data,
  output reg [3:0] q);
```

---

```verilog
    always @(posedge clk or posedge
    areset) begin
        if (areset) begin
            // Asynchronous positive edge
    triggered areset
            q <= 4'b0000;
        end else if (load) begin
            // Synchronous active high
    signals load
            q <= data;
        end else if (ena) begin
            // Shift right
            q <= {1'b0, q[3:1]};
        end
    end

endmodule
```

Correct Code

```verilog
    always @(posedge clk or posedge
     areset) begin
     if (areset) begin
        // Asynchronous positive edge
     triggered areset
        q <= 4'b0;
     end else if (load) begin
        // Synchronous active high signals
     load
        q <= data;
     end else if (ena) begin
        // Shift right
        q <= {q[2:0], 1'b0};
     end
    end

endmodule
```

Error Code

---

**LLM generated Error Report**

Error Type: shifting operation

Category: Sequential: shift registers

Description:
The error in the erroneous implementation is related to the shifting operation in the shift register.

```verilog
q <= {q[2:0], 1'b0};
```

This line is intended to shift the bits in the register to the right, but it incorrectly connects the least significant bit (q[0]) to the most significant bit (q[3]) instead of connecting a zero to the most significant bit. This results in a circular shift instead of a right shift.

To correct this error, the line should be changed to:

```verilog
q <= {1'b0, q[3:1]};
```

Here are the steps to repair the erroneous implementation
1. Identify the line of code that performs the shifting operation
2. Notice that the least significant bit (q[0]) is connected to the most significant bit (q[3]) instead of a zero.
3. Change the line of code to connect a zero to the most significant bit and shift the remaining bits to the right
4. Test the implementation to ensure that it behaves as expected.

---

---

**LLM constructed Repair data from open-source code**

**Input:**

You are provided with a Verilog module that implements a shift register. However, the implementation contains a common error related to shifting operations. Your task is to identify the error, which involves a mistake in the way bits are shifted within the register. Specifically, the problem results in an incorrect circular shift rather than the intended right shift operation. You need to fix the error to ensure that the register shifts correctly.

Erroneous Implementation:

```verilog
module shift_registers_0 (clk, clken, SI, SO);
    parameter WIDTH = 32;
    input clk, clken, SI;
    output SO;
    reg [WIDTH—1:0] shreg;
    always @(posedge clk)
    begin
        if (clken)
            // Erroneous shifting operation: circular shift instead of right shift
            shreg = {shreg[WIDTH—2:0], shreg[WIDTH—1]};
    end
    assign SO = shreg[WIDTH—1];
endmodule
```

In this erroneous implementation:
- The shift register incorrectly shifts the most significant bit (MSB) into the least significant bit (LSB), creating a circular shift.
- The correct behavior should be shifting in the new input bit ('SI') while moving existing bits to the right.

Hints for Fixing:
1. **Identify the shifting operation error:**
The line 'shreg = {shreg[WIDTH-2:0], shreg[WIDTH-1]};' causes a circular shift because the MSB is incorrectly fed back into the LSB position.
2. **Fix the shifting logic:**
Replace the erroneous MSB assignment with the correct new input bit ('SI'). The correct line should be 'shreg = {shreg[WIDTH-2:0], SI};' to shift in the new input bit from the 'SI' input and ensure the bits shift to the right.
By fixing the shifting operation, the register will now shift correctly, with the new input bit filling the LSB and the existing bits shifting to the right.

**Output:**

```verilog
module shift_registers_0 (clk, clken, SI, SO);
    parameter WIDTH = 32;
    input clk, clken, SI;
    output SO;
    reg [WIDTH—1:0] shreg;
    always @(posedge clk)
    begin
    if (clken)
        shreg = {shreg[WIDTH—2:0], SI};
    end
    assign SO = shreg[WIDTH—1];
endmodule
```

Figure 6: Another example based on *shift4* from VerilogEval-Human.

---

**Problem description:** *multi_booth_8bit* from RTLLM

Please act as a professional verilog designer.

Implement an 8-bit Radix-4 booth multiplier that performs the multiplication of two 8-bit inputs (a and b) using the Booth algorithm. It utilizes a clock signal (clk), and a reset signal (reset), and provides the product output (p) and a ready signal (rdy). The ready signal (rdy) is set to 1 to indicate the completion of the multiplication process.

Module name:
    multi_booth_8bit

Input ports:
    clk: Clock signal used for synchronous operation.
    reset: Reset signal used to initialize the multiplier module.
    a: 8-bit input representing the multiplicand.
    b: 8-bit input representing the multiplier.

Output ports:
    p: 16-bit output representing the product of the multiplication.
    rdy: Ready signal indicating the completion of the multiplication operation.

Implementation:
    On the positive edge of the clock signal (clk) or the positive edge of the reset signal (reset), the module performs the multiplication process.
    If the reset signal (reset) is high, two 16-bit registers multiplier <= 8a[7], a and multiplicand <= 8b[7], b.
    If the reset signal (reset) is low, indicating normal operation, the module checks if the counter (5bit ctr) is less than 16.
    If the counter (ctr) is less than 16, the multiplicand register (multiplicand) is left-shifted by 1 to simulate the Booth algorithm's shifting operation.
    If the current bit of the multiplier register (multiplier[ctr]) is 1, indicating a positive Booth encoding, the product register (p) accumulates the value of the multiplicand register (multiplicand).
    The counter (ctr) is incremented by 1. Once the counter (ctr) reaches 16, indicating the completion of the multiplication process, the ready signal (rdy) is set to 1.

Give me the complete code.

---

```verilog
module multi_booth_8bit (p, rdy, clk, reset, a, b);
  input clk, reset;
  input [7:0] a, b;
  output reg [15:0] p;
  output reg rdy;
  reg [15:0] multiplicand, multiplier;
  reg [4:0] ctr;
  always @(posedge clk or posedge reset) begin
    if (reset) begin
      multiplier <= {{8{a[7]}}, a};
      multiplicand <= {{8{b[7]}}, b};
      p <= 0;
      ctr <= 0;
      rdy <= 0;
    end else begin
      if (ctr < 16) begin
        multiplicand <= multiplicand << 1;
        if (multiplier[ctr]) begin
          p <= p + multiplicand;
        end
        ctr <= ctr + 1;
      end else begin
        rdy <= 1;
      end
    end
  end
endmodule
```
**Correct Code**

```verilog
module multi_booth_8bit (p, rdy, clk, reset, a, b);
  input clk, reset;
  input [7:0] a, b;
  output reg [15:0] p;
  output reg rdy;
  reg [15:0] multiplier, multiplicand;
  reg [4:0] ctr;
  always @(posedge clk or posedge reset) begin
    if (reset) begin
      multiplier <= {{8{a[7]}}, a};
      multiplicand <= {{8{b[7]}}, b};
      ctr <= 0;
      rdy <= 0;

    end else begin
      if (ctr < 16) begin
        multiplicand <= multiplicand << 1;
        if (multiplier[ctr])
          p <= p + multiplicand;
        ctr <= ctr + 1;
      end else begin
        rdy <= 1;
      end
    end
  end
endmodule
```
**Error Code**

---

**LLM generated Error Report**

Error Type: Incorrect initialization
Category: Sequential: counters
Description:
In the erroneous implementation, the p register is not initialized to zero during reset. This can lead to incorrect multiplication results, as the initial value of p is unknown and may contain random values. The correct implementation initializes the p register to zero during reset, ensuring that the multiplication result is accurate.

To repair the erroneous implementation, follow these steps:
1. Locate the always block that handles the reset and clock edge. In this case, it's the block that starts with always @(posedge clk or posedge reset).
2. Within the if (reset) block, add a line to initialize the p register to zero: p <= 0;.

---

**LLM constructed Repair data from open-source code**

**Input:**

Problem Description:
You are given a Verilog module that implements a 4-bit up counter. The counter should reset to zero when the reset signal is active and increment with each positive clock edge. However, there is a common error in this implementation: the counter_up register is not correctly initialized during reset, which may lead to unpredictable behavior after reset. Your task is to identify and fix this initialization error. Erroneous Implementation:

```
module up_counter(
    input clk,
    input reset,
    output [3:0] counter
    );
    reg [3:0] counter_up;
    // up counter
    always @(posedge clk or posedge reset)
    begin
        if (reset)
            counter_up <= 4'd3;  // Incorrect initialization value
        else
            counter_up <= counter_up + 4'd1;
    end
    assign counter = counter_up;
endmodule
```

In this erroneous implementation:
* The counter_up register is incorrectly initialized to 4'd3 during reset, instead of being reset to zero (4'd0), which can cause incorrect counter behavior when reset is activated.

Hints for Fixing:
1. Identify the initialization issue: The counter should reset to zero, but currently, it is being set to 4'd3 during reset. This causes incorrect counting behavior after a reset.
2. Fix the reset logic: Modify the if (reset) block in the always block to properly initialize the counter to zero. Change counter_up <= 4'd3; to counter_up <= 4'd0;.
3. Ensure proper reset behavior: After fixing the initialization, the counter will correctly reset to zero when the reset signal is active and increment correctly on each clock cycle thereafter.

**Output:**

```
module up_counter(
    input clk,
    input reset,
    output[3:0] counter
    );
    reg [3:0] counter_up;
    // up counter
    always @(posedge clk or posedge reset)
    begin
        if(reset)
            counter_up <= 4'd0;
        else
            counter_up <= counter_up + 4'd1;
    end
    assign counter = counter_up;
endmodule
```

Figure 7: Another example based on *multi_booth_8bit* from RTLLM.

# D EXAMPLES OF CORRECT-BY-CONSTRUCTION DATA FOR NON-TEXTUAL REPRESENTATIONS

## D.1 KARNAUGH MAPS AND TRUTH TABLES

**Problem**

```
Implement the circuit described by the Karnaugh map below.
//      c
// ab   0   1
// 00 | 1 | 0
// 01 | 0 | 0
// 11 | 0 | 0
// 10 | 0 | 0

module top_module(
        input a,
        input b,
        input c,
        output out
);
```

**Solution**

```
The input variables are: ['a', 'b', 'c'].
Based on the Karnaugh map, I can transform in to the following truth table:
 a | b | c | f
 0 | 0 | 0 | 1
 0 | 0 | 1 | 0
 0 | 1 | 0 | 0
 0 | 1 | 1 | 0
 1 | 0 | 0 | 0
 1 | 0 | 1 | 0
 1 | 1 | 0 | 0
 1 | 1 | 1 | 0

The minterms (when output is 1) are:
(0,0,0) => (~a & ~b & ~c)
This corresponds to the following minterms logic:
'(~a & ~b & ~c)'

Finally, based on the above logic equation, I can now write the Verilog code that
    could be described by the Karnaugh map:
'''
module top_module(
        input a,
        input b,
        input c,
        output out
);
        assign out = (~a & ~b & ~c);
endmodule
'''
```

Figure 8: Example of a generated problem with Karnaugh Map representation in the problem description. We include detailed step-by-step analysis for reaching the correct solution.

## D.2    STATE TRANSITION GRAPHS AND TABLES

**Problem**

```
Given the state-assigned table shown below, implement the logic functions Y[0] and
    z.
// Present state y[2:0] | Next state Y[2:0] x=0, Next state Y[2:0] x=1 | Output z
// 000 | 010, 011 | 1
// 001 | 100, 010 | 0
// 010 | 001, 100 | 1
// 011 | 011, 100 | 0
// 100 | 100, 001 | 0
```

**Solution**

```
The state transition is as follows:
// state | Next state in=0, Next state in=1 | Output
// A | C, D | 1
// B | E, C | 0
// C | B, E | 1
// D | D, E | 0
// E | E, B | 0

The transition logic is then:
A: next = x ? D : C;
B: next = x ? C : E;
C: next = x ? E : B;
D: next = x ? E : D;
E: next = x ? B : E;

The output is 1 for states: A, C.
Thus the output logic is: assign z = ( y == A || y == C );
Y0 corresponds to 001 (A), 011 (D).

Finally, below is the Verilog code for the finite state machine:
'''
module top_module (
 input clk,
 input x,
 input [2:0] y,
 output reg Y0,
 output reg z
);
        reg [2:0] next_state;
        parameter A=0, B=1, C=2, D=3, E=4;
        always_comb begin
                case(y)

                        A: next_state = x ? D : C;
                        B: next_state = x ? C : E;
                        C: next_state = x ? E : B;
                        D: next_state = x ? E : D;
                        E: next_state = x ? B : E;
                        default: next_state = 'x;
                endcase
        end
        assign z = ( y == A || y == C );
        assign Y0 = ( next_state == B || next_state == D );
endmodule
'''
```

Figure 9: Example of problem with state transition tables.

**Problem**

```
This is a Moore state machine with four states, four inputs, and one output.
    Implement this state machine in Verilog. Reset is an active—high synchronous
    reset to state D.

// D (out=0) —in0=1—> D
// D (out=0) —in0=0—> A
// C (out=0) —in1=1—> D
// C (out=0) —in1=0—> B
// B (out=1) —in2=1—> D
// B (out=1) —in2=0—> B
// A (out=0) —in3=1—> B
// A (out=0) —in3=0—> C
```

**Solution**

```
The finite state machine has four inputs, and the state transition logic is as
    follows:
D: next = in0 ? D : A;
C: next = in1 ? D : B;
B: next = in2 ? D : B;
A: next = in3 ? B : C;

The output is 1 for states: B.
Thus the output logic is: 'assign out = ( state == B );'.

Finally, below is the Verilog code for the finite state machine:
'''
module top_module (
 input clk,
 input reset,
 input in0,
 input in1,
 input in2,
 input in3,
 output out
);
        parameter D=0, C=1, B=2, A=3;
        reg state;
        reg next;
        always_comb begin
                case(state)
                        D: next = in0 ? D : A;
                        C: next = in1 ? D : B;
                        B: next = in2 ? D : B;
                        A: next = in3 ? B : C;
                        default: next = 'x;
                endcase
        end
        always @(posedge clk) begin
                if (reset) state <= D;
                else state <= next;
        end
        assign out = ( state == B );
endmodule
'''
```

Figure 10: Example of problem with Moore FSM.

**Problem**

The following diagram is a Mealy machine. Implement in Verilog using one—hot
encoding. Resets into state A and reset is asynchronous active—high.

```
// A —x=0 (z=0)—> D
// A —x=1 (z=1)—> C
// B —x=0 (z=1)—> C
// B —x=1 (z=0)—> B
// C —x=0 (z=0)—> C
// C —x=1 (z=0)—> D
// D —x=0 (z=1)—> C
// D —x=1 (z=0)—> B
```

**Solution**

From the transition diagram, we have the following transition logic:

```
// state | next state in=0, next state in=1
// A | D, C
// B | C, B
// C | C, D
// D | C, B
```

Thus the state transition logic is as follows:

```
A: next = x ? C : D;
B: next = x ? B : C;
C: next = x ? D : C;
D: next = x ? B : C;
```

The output is 1 for states: (A, x), (B, ~x), (D, ~x).
Thus the output logic is: 'assign z = ( ( state == A & x ) || ( state == B & ~x )
|| ( state == D & ~x ) );'.

Finally, below is the Verilog code for the finite state machine:
```
module top_module (
        input clk,
        input areset,
        input x,
        output z
);
        parameter A=2'b00, B=2'b01, C=2'b10, D=2'b11;
        reg [1:0] state;
        reg [1:0] next_state;
        always_comb begin
                case(state)

                        A: next_state = x ? C : D;
                        B: next_state = x ? B : C;
                        C: next_state = x ? D : C;
                        D: next_state = x ? B : C;
                        default: next_state = 'x;
                endcase
        end
        always @(posedge clk, posedge areset) begin
                if (areset) state <= A;
                else state <= next_state;
        end
        assign z = ( ( state == A & x ) || ( state == B & ~x ) || ( state == D & ~x
    ) );
endmodule
```

Figure 11: Example of problem with Mealy FSM.

**Problem**

```
The following is the state transition table for a Moore state machine with one
    input, one output, and four states. Use the following one-hot state encoding: A
    =4'b0001, B=4'b0010, C=4'b0100, D=4'b1000. Derive state transition and output
    logic equations by inspection assuming a one-hot encoding. Implement only the
    state transition logic and output logic (the combinational logic portion) for
    this state machine.
// state | Next state in=0, Next state in=1 | Output
// A | B, A | 0
// B | B, C | 1
// C | D, A | 1
// D | B, C | 0
```

**Solution**

```
Based on the state transition table, we can obtain the next state from observing
    the row (previous state) and column (input).
Next state is A on the following (row, column): (A, in=1) (C, in=1). This
    correspond to the following logic: 'state[A] & in || state[C] & in'.
Next state is B on the following (row, column): (A, in=0) (B, in=0) (D, in=0). This
     correspond to the following logic: 'state[A] & ~in || state[B] & ~in || state[
    D] & ~in'.
Next state is C on the following (row, column): (B, in=1) (D, in=1). This
    correspond to the following logic: 'state[B] & in || state[D] & in'.
Next state is D on the following (row, column): (C, in=0). This correspond to the
    following logic: 'state[C] & ~in'.

The output is 1 for states: B, C.
Thus the output logic is: 'assign out = ( state[B] || state[C] );'.

Finally, below is the Verilog code for the finite state machine:
'''
module top_module (
 input in,
 input [3:0] state,
 output reg [3:0] next_state,
 output out
);

        parameter A=0, B=1, C=2, D=3;

        assign next_state[A] = state[A] & in || state[C] & in;
        assign next_state[B] = state[A] & ~in || state[B] & ~in || state[D] & ~in;
        assign next_state[C] = state[B] & in || state[D] & in;
        assign next_state[D] = state[C] & ~in;

        assign out = ( state[B] || state[C] );

endmodule
'''
```

Figure 12: Example of problem with in-edge focused solution shown in Figure 2.

## D.3 WAVEFORMS

---

**Problem**

This is a combinational circuit. Read the simulation waveforms to determine what
    the circuit does, then implement it.

```
// time    a         b         c         d         q
// 0ns     0         0         0         0         0
// 5ns     0         0         0         0         0
// 10ns    0         0         0         0         0
// 15ns    0         0         0         0         0
// 20ns    0         0         0         1         0
// 25ns    0         0         1         0         0
// 30ns    0         0         1         1         0
// 35ns    0         1         0         0         0
// 40ns    0         1         0         1         0
// 45ns    0         1         1         0         0
// 50ns    0         1         1         1         0
// 55ns    1         0         0         0         1
// 60ns    1         0         0         1         1
// 65ns    1         0         1         0         0
// 70ns    1         0         1         1         1
// 75ns    1         1         0         0         0
// 80ns    1         1         0         1         1
// 85ns    1         1         1         0         0
// 90ns    1         1         1         1         0
```

**Solution**

Based on the simulation waveform, I can transform in to the following truth table:

```
a | b | c | d | f
0 | 0 | 0 | 0 | 0
0 | 0 | 0 | 1 | 0
0 | 0 | 1 | 0 | 0
0 | 0 | 1 | 1 | 0
0 | 1 | 0 | 0 | 0
0 | 1 | 0 | 1 | 0
0 | 1 | 1 | 0 | 0
0 | 1 | 1 | 1 | 0
1 | 0 | 0 | 0 | 1
1 | 0 | 0 | 1 | 1
1 | 0 | 1 | 0 | 0
1 | 0 | 1 | 1 | 1
1 | 1 | 0 | 0 | 0
1 | 1 | 0 | 1 | 1
1 | 1 | 1 | 0 | 0
1 | 1 | 1 | 1 | 0
```

The minterms (when output is 1) are:
(1,0,0,0) => (a & ~b & ~c & ~d)
(1,0,0,1) =>  (a & ~b & ~c & d)
(1,0,1,1) => (a & ~b & c & d)
(1,1,0,1) =>  (a & b & ~c & d)
This corresponds to the following minterms logic:
'(a & ~b & ~c & ~d) | (a & ~b & ~c & d) | (a & ~b & c & d) | (a & b & ~c & d)'

Finally, based on the above logic equation, I can now write the Verilog code:
'''

```
module top_module(
        input a,
        input b,
```

```
        input c,
        input d,
        output q
);
        assign q = (a & ~b & ~c & ~d) | (a & ~b & ~c & d) | (a & ~b & c & d) | (a &
    b & ~c & d);
endmodule
'''
```

Figure 13: Example of a waveform problem for combinational circuits.

**Problem**

```
This is a sequential circuit. Read the simulation waveforms to determine what the
    circuit does, then implement it.

// time          clk           reset         in            out
// 0ns           0             1             0             x
// 5ns           1             1             0             1
// 10ns          0             1             0             1
// 15ns          1             0             0             1
// 20ns          0             0             0             1
// 25ns          1             0             1             0
// 30ns          0             0             1             0
// 35ns          1             0             0             1
// 40ns          0             0             0             1
// 45ns          1             0             0             0
// 50ns          0             0             0             0
// 55ns          1             0             0             1
// 60ns          0             0             0             1
// 65ns          1             0             1             0
// 70ns          0             0             1             0
// 75ns          1             0             1             1
// 80ns          0             0             1             1
// 85ns          1             0             0             1
// 90ns          0             0             1             1
```

... Simulation waveform redacted ...

**Solution**

```
From the waveform, we have the following transition logic and output logic:
// state | Next state in=0, Next state in=1 | Output
// D | C, B | 1
// C | B, D | 0
// B | D, A | 1
// A | C, B | 0

Thus the state transition logic is as follows:
D: next = in ? B : C;
C: next = in ? D : B;
B: next = in ? A : D;
A: next = in ? B : C;

The output is 1 for states: D, B.
Thus the output logic is: 'assign out = ( state == D || state == B );'.

Finally, below is the Verilog code for the finite state machine:
```

```
'''
module top_module (
 input clk,
 input in,
 input reset,
 output out
);

        parameter D=0, C=1, B=2, A=3;
        reg state;
        reg next;

        always_comb begin
                case(state)

                        D: next = in ? B : C;
                        C: next = in ? D : B;
                        B: next = in ? A : D;
                        A: next = in ? B : C;
                        default: next = 'x;
                endcase
        end

        always @(posedge clk) begin
                if (reset) state <= D;
                else state <= next;
        end

        assign out = ( state == D || state == B );

endmodule
'''
```

Figure 14: Example of a waveform problem for sequential circuits.

# E  PROMPT TEMPLATES

## E.1  SYNTHETIC DATA GENERATION

### E.1.1  SELF-INSTRUCT

---

Your goal is to create a high-quality Verilog problem.

* Guidelines for designing the problem description:

1. This should be **completely self-contained**, providing all the contextual information one needs to understand and solve the problem.
2. Assume common verilog knowledge, but ensure that any specific context, variables, or code snippets pertinent to this problem are explicitly included.
3. Do not include the code snippet in the problem.
4. The problem should be desinged for the programmers to solve it with one verilog module.
5. The problem description section should be enclosed within <PROBLEM> </PROBLEM> tags.

Now, Please use your creativity to create a brand new high-quality Verilog problem.

---

Figure 15: Prompt used to generate initial 50 seed problems for **Self-Instruct**.

---

Your goal is to create a high-quality Verilog problem.

* Guidelines for designing the problem description:

1. This should be **completely self-contained**, providing all the contextual information one needs to understand and solve the problem.
2. Assume common verilog knowledge, but ensure that any specific context, variables, or code snippets pertinent to this problem are explicitly included.
3. Do not include the code snippet in the problem.
4. The problem should be desinged for the programmers to solve it with one verilog module.
5. The problem description section should be enclosed within <PROBLEM> </PROBLEM> tags.

Below shows some examples:

<PROBLEM>
{seed problems}
</PROBLEM>

Now, Please use your creativity to create a brand new high-quality Verilog problem.

---

Figure 16: Prompt used for **Self-Instruct**.

### E.1.2 OSS-INSTRUCT

Your goal is to create a high-quality Verilog problem.

* Guidelines for designing the problem description:

1. This should be **completely self-contained**, providing all the contextual information one needs to understand and solve the problem.
2. Assume common verilog knowledge, but ensure that any specific context, variables, or code snippets pertinent to this problem are explicitly included.
3. Do not include the code snippet in the problem.
4. The problem should be designed for the programmers to solve it with one Verilog module.

* Guidelines for the problem description format: The problem description section should be enclosed within <PROBLEM> </PROBLEM> tags.

Please increase the difficulty of the given programming test question a bit. You can increase the difficulty using, but not limited to, the following methods:

1. Your new problem should not be directly solved by the original code snippet.
2. You can also change the bit-width requiremnt, how to reset internal signals (if applicable), and whether the solution needs a clock signal (combinatorial versus sequential logic). If you do have a reset method that is synchronous to a clock, make sure to add the clock signal to the problem module input.
3. Add new constraints and requirements to the original problem, adding approximately 10 additional words.
4. Replace a commonly used requirement in the programming task with a less common and more specific one.
5. If the original problem can be solved with only a few logical steps, please add more reasoning steps.

Now, Please gain inspiration from the following random code snippet to create a high-quality Verilog problem.

Code snippet for inspiration:
```
{code snippet}
```

Output:

Figure 17: Prompt used for **OSS-Instruct**. We also include prompts inspired from Evol-Instruct (Luo et al., 2024b) to increase problem difficulty.

### E.1.3 DOCU-INSTRUCT

Your goal is to create a high-quality Verilog problem.

* Guidelines for designing the problem description:

1. This should be \*\*completely self-contained\*\*, providing all the contextual information one needs to understand and solve the problem.
2. Assume common verilog knowledge, but ensure that any specific context, variables, or code snippets pertinent to this problem are explicitly included.
3. Do not include the code snippet in the problem.
4. The problem should be designed for the programmers to solve it with one Verilog module.

* Guidelines for the problem description format: The problem description section should be enclosed within <PROBLEM> </PROBLEM> tags.

Now, Please gain inspiration from the following textbook or wikipedia snippet to create a high-quality Verilog problem. The information might not be directly related to Verilog, but try to be make the problem as relevant as possible to the textbook issue discussed.

Textbook snippet for inspiration:
```
{document snippet}
```

Output:

Figure 18: Prompt used for **Docu-Instruct** with Wikipedia and textbooks.

I am going to give you a concept and some descriptions about that concept. Based on the descriptions and concept name, determine if the concept belongs to one of the following categories:

- Hardware description and modeling in Verilog.
- Fundamental constructs such as modules, ports, and wires specific to Verilog.
- Synthesis and optimization techniques employed in hardware design using Verilog.
- Simulation tools and methodologies for verifying Verilog-based hardware designs.
- Common design patterns and best practices in Verilog for efficient hardware implementation.
- Programming concepts like loops, functions related to Verilog.
- Hardware related concepts such as finite state machines that could be implemented in Verilog.
- Algorithms that could be implemented in hardware, such as Fourier Transforms.

Concept: {Wikipedia title}
Description: {Wikipedia content}

Do not make assumptions and only respond "Yes" if you are certain that the {Wikipedia title} is related to hardware design or Verilog coding language.

Your answer should start with "Yes" or "No".

Figure 19: Prompt used to filter Verilog related Wikipedia pages.

### E.1.4 NON-TEXTUAL REPRESENTATIONS

Your goal is to create a high-quality Verilog problem. Specifically, we would like to test the skills of understanding Karnaugh maps and state transition diagrams. The problem description section should be enclosed within <PROBLEM> </PROBLEM> tags.

Now, please gain inspiration from the following random code snippet to create a high-quality Verilog problem. Remember that the problem you generated must include Karnaugh maps in the format above. The random code snippet MUST be related to the solution. Your problem statement should be short and succinct (no more than 5 sentences) and you MUST generate a Karnaugh map in the problem description. Your problem description should not describe the Karnaugh map in words and should assume that the student need to decipher the Karnaugh map to solve the problem.

Code snippet for inspiration:
```
{code snippet}
```

Below are two examples on how to represent Karnaugh map related questions in purely textual format. You should NOT use the following to generate the problem but only consider the style.

```
<PROBLEM>
Given the state-assigned table shown below, implement the finite-state machine.
    Reset should synchronous active high reset the FSM to state 000.
// Present state y[2:0] | Next state y[2:0] x=0, Next state y[2:0] x=1, Output z
// 000 | 000, 001 | 0
// 001 | 001, 100 | 0
// 010 | 010, 001 | 0
// 011 | 001, 010 | 1
// 100 | 011, 100 | 1
</PROBLEM>
<PROBLEM>
Implement the circuit described by the Karnaugh map below.
//        a
// bc    0 1
//  00 | 0 | 1 |
//  01 | 1 | 1 |
//  11 | 1 | 1 |
//  10 | 1 | 1 |
</PROBLEM>
```

Figure 20: An prompt example to encourage LLMs to generate questions with Karnaugh Maps.

### E.1.5 PROMPTS FOR SAMPLING SOLUTIONS WITH LLM GENERATED PROBLEMS

Your goal is to solve a Verilog problem by completing one verilog module.

* Guidelines for writing the solution (verilog module) and formatting:
1. The module should be **completely self-contained**, fulfilling all the requirements needed to solve the problem.
2. Ensure the correctness of the syntax and functionality.
3. The module name must be "top_module".
4. Do not add blackslash in front of underscore symbol.
5. The solution must be valid Verilog code.
6. The solution section should be enclosed within <SOLUTION> </SOLUTION> tags.
7. The reasoning section should be enclosed within <REASON> </REASON> tags.
8. Write comments in the solution section. Try to embed the reasoning in comments. Always try to write the corresponding comments before the code.
9. The module should be **completely self-contained**, do not include or import outside the module and define everything inside the module.

Below shows an example:

Problem description:

```
Build a counter that counts from 0 to 999, inclusive, with a period of 1000 cycles.
    The reset input is active high synchronous, and should reset the counter to 0.
    Solve the problem by completing the following module.
```

Output:

```
<REASON>
To solve the problem of building a counter that counts from 0 to 999 with a reset
    input that is active high synchronous, you will need a 10—bit register (since
    2^10=1024 and 999 < 1024). The counter should increment on each clock cycle and
     wrap around back to 0 when it reaches 1000. The reset input, when active,
    should reset the counter to 0 on the next clock edge.
</REASON>
```

```verilog
Here is the Verilog code for the module:
<SOLUTION>
module top_module(
    input clk,
    input reset,
    output reg [9:0] q);

    // This block executes on the positive edge of the clock signal.
    always @(posedge clk) begin
        // Checks if the reset is active. This is synchronous with the clock.
        if (reset) begin
            // Reset the counter to 0 when reset is active
            q <= 0;
        // If the counter has reached 999, it needs to wrap around to 0 on the next
    cycle.
        end else if (q == 999) begin
            // Reset to 0 after reaching 999
            q <= 0;
        // In all other cases, just increment the counter.
        end else begin
            // Increment the counter
            q <= q + 1;
        end
    end
endmodule
</SOLUTION>
```

Now, please solve the following Verilog problem. I will also attach a reference code snippet which was used as an inspiration to generate the problem. The provided code may not directly solve the problem so you should use it only as a reference.

Reference code:
```
{code snippet}
```

Problem description:
```
{in context examples}
```
Output:

Figure 21: Prompt used for sampling solutions for synthetic data generation. We include a in context example to encourage models to include reasoning traces. Prompts in blue are only included for problems generated from a code snippet.

### E.1.6 PROMPTS FOR VERIFYING SOLUTIONS

Check if the given Verilog module is a valid solution to the problem. The output should be in "True" or "False" and be enclosed within <VALID> </VALID> tags and the explanation in <REA-SON></REASON> tags.

Now check the following:

<PROBLEM>
{problem}
<PROBLEM>

<SOLUTION>
{solution}
</SOLUTION>

Figure 22: Prompt used for verifying solutions.

### E.2 PROMPTS FOR TARGETED CODE REPAIR

### E.2.1 ERROR REPORT

```
Here is a Verilog problem description:
```
{problem description}
```

Here is an erroneous implementation:
```
{error code}
```

Here is a correct implementation:
```
{correct code}
```

Generate a detail error report.
The error report should describe the common error type and output the code category. The error report should also be detailed enough to let beginners to repair the erroneous implementation step by step.

Output:
```

Figure 23: Prompt for **Error Report** generation.

```
Here is a Verilog problem description:
```
{problem description}
```

Here is an erroneous implementation:
```
{error code}
```

Here is the error report:
```
{error report}
```

Now fix the erroneous implementation and give me the correct code.

Output:
```

Figure 24: Prompt for **Error Report** self-consistency validation. The generated code fix will be evaluated for functional correctness. Error reports whose code fixes do not pass will be filtered.

E.2.2  ERROR INJECTION

Your goal is to create an error-fixing Verilog practice problem for programmers. You will demonstrate a type of error that is commonly made by programmers.
Create an error repair practice problem with three components:
1. Problem description
2. Erroneous implementation
3. Hints for fixing

Here is an example:

<EXAMPLE>
The following Verilog module is intended to implement the specification below. However, there is a bug in the code which causes incorrect results. Please fix the bug to make the module work as intended.

Erroneous Implementation:

```
// Verilog code with the injected error
module example_module (
    input wire clk,
    input wire reset,
    output reg [3:0] counter
);

// Intended functionality:
// This module should count from 0 to 15 and then wrap around.

always @(posedge clk or posedge reset) begin
    if (reset) begin
        counter <= 4'b0000;
    end else begin
        counter <= counter + 1'b1; // Error injected: Should be 4'b1
    end
end

endmodule
```

Hints for Fixing:
1. Verify the bit-width of the counter and the increment operation.
2. Check the initialization and wrapping condition of the counter.
3. Ensure that the addition operation correctly handles the 4-bit counter.

</EXAMPLE>

Now, here is the commonly made error:
```
{error report}
```

Inject the above error into the following module and create an error repair practice problem. Check if it is possible to inject the error. If not, create the problem with the given error alone and ignore the module in the code snippet.

```
{code snippet}
```

Output:

Figure 25: Prompt used to inject targeted errors to open-source code in code **Repair** data. We also prompt the LLM to self-verify if the error could be injected to the code snippet.

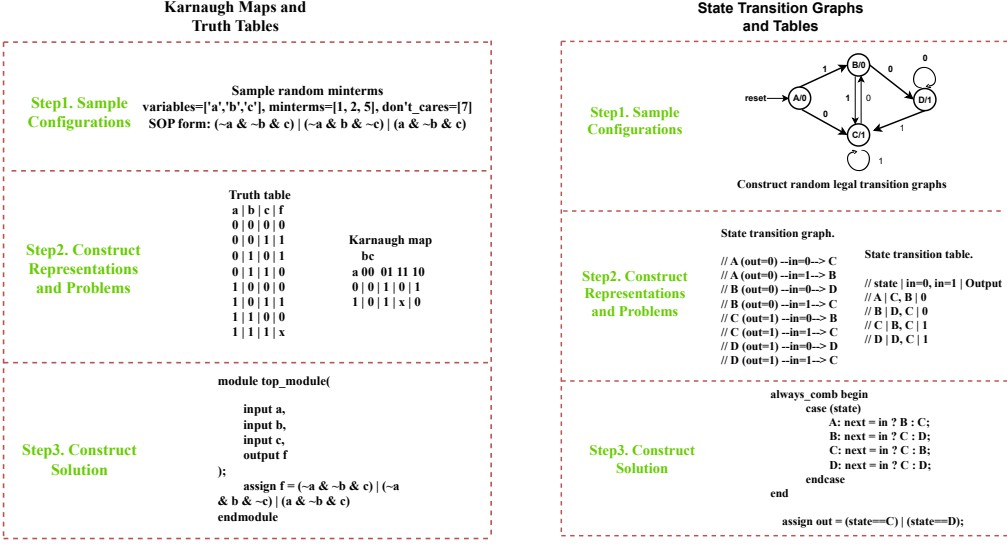

Figure 26: Correct-by-construction for Karnaugh maps and truth tables.

Figure 27: Correct-by-construction for finite-state machines.

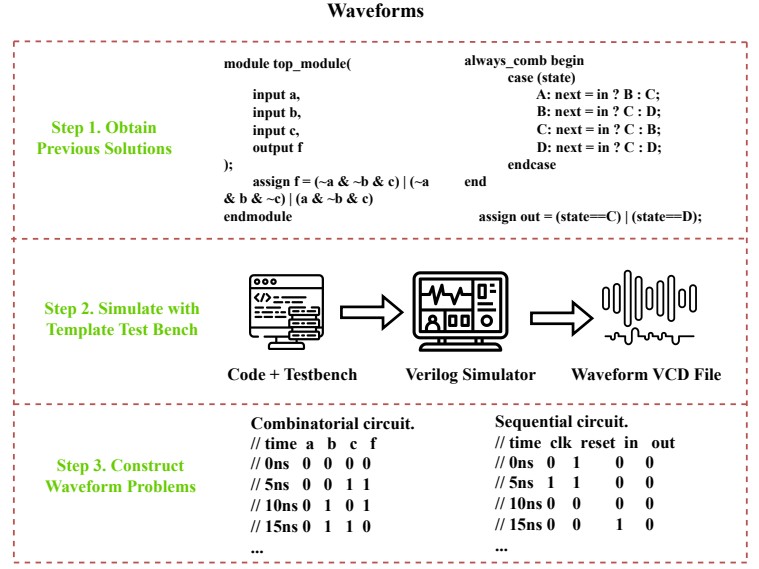

Figure 28: Correct-by-construction for waveforms.

