# OpenReview forum: "CraftRTL: High-quality Synthetic Data Generation for Verilog Code Models with Correct-by-Construction Non-Textual Representations and Targeted Code Repair"
_ICLR.cc/2025/Conference — ICLR 2025 Poster_

### Official Review · Reviewer_uCA7 · 2024-10-21

**Soundness:** 4
**Presentation:** 4
**Contribution:** 3
**Rating:** 8
**Confidence:** 3

**Summary:**

- This paper performs a thorough analysis of fine-tuned LLMs on Verilog code, and revealing two main challenges of automated Verilog code generation
- This paper creates a large number of correct-by-construction data to ensure solution correctness, incorporating Karnaugh Maps, state-transition diagrams, and waveforms
- This paper develops an automated framework that utilizes LLMs to generate error reports from benchmark problems
- Its evaluation results demonstrate that models fine-tuned with our data achieve state-of-the-art performance on Verilog coding, outperforming prior SOTA results by 3.8%, 10.9%, 6.6% for pass@1 on VerilogEval-Machine, VerilogEval-Human, and RTLLM, respectively

**Strengths:**

- This paper is clearly written and easy to comprehend
- This paper is well-motivated and address an important downstream task, automated Verilog code generation
- This paper includes a comprehensive and reliable data construction pipeline
- This paper conduct a comprehensive evaluation on three LLMs with SOTA baselines

**Weaknesses:**

Actually I like this paper, especially the data construction section; however, there are still some minor concerns:

- "Quality Assurance with LLM Validation" (Line 317): Please provide more evidence about the choice of LLM validation. What is the rationale (or its limitations) of not using deterministic validation approaches, e.g., model checking?
- In Section 2.3, you mention "significant variability in the model’s pass rate on specific benchmark problems across different checkpoints" (Line 164), while the results in Figure 1 indicates a highly positive correlation. Also, 15% discrepancies is also acceptable between two checkpoints. Can you provide me with a stronger evidence to support this claim, e.g., Pearson Correlation Coefficient, or explain why such difference is significant in this task?
- Application scenario: This paper mainly utilizes domain specific patterns of various types of Verilog while it might be difficult when applied to similar tasks, e.g., code generation without sufficient training data.
- Reproducibility Statement: this subsection exceeds the 10 pages limit. I think it should be placed within the first 10 pages or directly moved to appendix
- Availability: this paper does not provide an available artifact

**Questions:**

- Checkpoint Selection: what is the selection criteria of your checkpoints in Figure 1 and Figure 5? You mentions "two consecutive checkpoints" in Line 434. Additionally, you only fine-tune your model for one epoch (Line 364), so at least one checkpoint might not see all training data. Whether such difference affects your results?
- Application Scenario: This paper addresses an important problem in Verilog code generation utilizing domain knowledge of Verilog. So I am curious how such approach is applied to similar tasks, e.g., code generation without sufficient training data?

---

> ### Author Response · Authors · 2024-11-19
>
> Thank you for your review and recognition of our work. We provide the following comments regarding Weaknesses and answers to Questions.
>
> *W1. "Quality Assurance with LLM Validation" (Line 317): Please provide more evidence about the choice of LLM validation. What is the rationale (or its limitations) of not using deterministic validation approaches, e.g., model checking?*
>
> AW1.  The model we used for the self-consistency check, nemotron-340b-instruct, is weaker on VerilogEval than models used to generate correct/error code (CC etc. models). It is largely ineffective at correcting mistakes without proper guidance from error reports. To validate this, we prompt LLM to fix error code without error reports and obtain the fix rate of 13.3% (with error report should be 100%). The significant difference strongly emphasizes the importance of providing high-quality error reports to mitigate and fix the errors. We have updated our paper on Page 6 Line 313: “whereas directly prompting the LLM without detailed error reports could resolve only 13% of the errors”.
>
> We do not consider deterministic methods, such as formal verification for evaluating solution correctness. Formal verification would still require designers (or LLMs) to construct functional properties (System Verilog Assertions) from problem specifications. LLMs are still yet not fully capable to do so effectively, and improving LLMs for formal verification is an active research area [1,2].
>
> *W2. In Section 2.3, you mention "significant variability in the model’s pass rate on specific benchmark problems across different checkpoints" (Line 164), while the results in Figure 1 indicates a highly positive correlation. Also, 15% discrepancies is also acceptable between two checkpoints. Can you provide me with a stronger evidence to support this claim, e.g., Pearson Correlation Coefficient, or explain why such difference is significant in this task?*
>
> AW2. We have provided the Pearson Correlation Coefficient in Figure 1.  We have updated Figure 1 and provided further information in Appendix A10. We choose checkpoint2 to be the last checkpoint during training and checkpoint1 to be the immediate predecessor (64 gradient steps). The ideal outcome is not merely reduced variability but also less degradations and improved accuracy: specifically, most problems in checkpoint2 should show higher pass rates than checkpoint1, assuming that training on additional data enhances model performance. We select such a representation hoping to give readers an overall impression on training variability across all problems with two checkpoints. In Table 17 of Appendix A10 we present an alternative option of displaying the pass rates for selected benchmark problems throughout the training progression.
>
> *W3. Application scenario: This paper mainly utilizes domain specific patterns of various types of Verilog while it might be difficult when applied to similar tasks, e.g., code generation without sufficient training data.*
>
> AW3. Although our work is focused on the narrow domain of Verilog coding, we believe that our proposed methods could be generalizable (details in Appendix B), and be of value to the broad ICLR research community.
>
> *W4. Reproducibility Statement: this subsection exceeds the 10 pages limit. I think it should be placed within the first 10 pages or directly moved to appendix*
>
> AW4. We have modified our manuscript such that the Reproducibility Statement is within page limit.
>
> *W5. Availability: this paper does not provide an available artifact*
>
> AW5. We apologize for not making our research artifact available at time of review. To enhance reproducibility, we are committed to release the source code of our data generation pipeline.

---

> > ### Author Response · Authors · 2024-11-19
> >
> > *Q1. Checkpoint Selection: what is the selection criteria of your checkpoints in Figure 1 and Figure 5? You mention "two consecutive checkpoints" in Line 434. Additionally, you only fine-tune your model for one epoch (Line 364), so at least one checkpoint might not see all training data. Whether such a difference affects your results?*
> >
> > A1. We have provided further information in Appendix A10 regarding checkpoint selection. We only fine-tuned models for one epoch so you are absolutely correct that checkpoint2 would see more data than checkpoint1. The ideal outcome is not merely reduced variability but also less degradations and improved accuracy: specifically, most problems in checkpoint2 should show higher pass rates than checkpoint1, assuming that training on additional data enhances model performance. However, as shown in Figure 1a training on SDG data results in a significant degradation of pass rates for many problems between checkpoint1 and checkpoint2. In contrast Figure 1b demonstrates reduced degradation and improvement in more problems. We further elaborate such findings in Table 17 (Appendix A10), where we display pass rates for selected benchmark problems with high volatility from VerilogEval-Human throughout the training progression.
> >
> > *Q2.Application Scenario: This paper addresses an important problem in Verilog code generation utilizing domain knowledge of Verilog. So I am curious how such an approach is applied to similar tasks, e.g., code generation without sufficient training data?*
> >
> > A2. We further discuss in Appendix B how our approach is inherently adaptable to other HDLs and programming languages. In short, leveraging custom-designed solvers to generate accurate execution-based solutions is a versatile method applicable to any programming language. While this work focuses on Verilog, it is not limited to it and can be extended to various domain-specific languages (DSLs) for hardware design. This adaptability enables the pipeline to effectively address language-specific challenges while remaining useful across diverse domains. Additionally, to tackle code generation with limited training data, we emphasize that the error-repair approach can serve as a general solution. By identifying deficiencies in the model, we can better determine what data to collect, synthesize, or annotate to enhance the model’s capabilities in subsequent training iterations.
> >
> > *References*
> >
> > [1] Kang et al, "FVEval: Understanding Language Model Capabilities in Formal Verification of Digital Hardware"
> >
> > [2] Qayyum et al, "LLM-assisted Automated Incremental Proof Generation for Hardware Verification"

---

> > > ### Comment · Reviewer_uCA7 · 2024-11-19
> > >
> > > Thanks for providing the aforementioned explanation and addressing the concerns.

---

> > > > ### Author Response · Authors · 2024-11-21
> > > >
> > > > Thank you once more for your thoughtful feedback and acknowledgment of our efforts.

---

### Official Review · Reviewer_sdjN · 2024-10-27

**Soundness:** 3
**Presentation:** 2
**Contribution:** 3
**Rating:** 6
**Confidence:** 4

**Summary:**

This paper discusses two main issues when LLMs  handling Verilog code: models have difficulty handling non-textual elements in problem statements and models making "minor" programming mistakes. To address these issues, the authors specifically created a transformed non-textual dataset and code repair dataset to fine-tune the model. The results demonstrate that the fine-tuned Starcoder2-15B surpasses the prior state-of-the-art results in Pass@1 performance, achieving improvements of 3.8\%, 10.9\%, and 6.6\% on VerilogEval-Machine, VerilogEval-Human, and RTLLM, respectively.

**Strengths:**

1. The paper conducts a detailed empirical analysis of the two main issues in Verilog code.
2. The paper provides a thorough comparison with existing methods and shows good performance.

**Weaknesses:**

1. The main contribution of this work lies in constructing a fine-tuning dataset to address non-textual data and minor error issues. The technical contribution of the paper is limited.
2. What is the specific definition of "minor" errors, and what common characteristics do they share?
3. The font size of Figures 2 and 4 is too small to read.

**Questions:**

1. Why focus solely on karnaugh maps, state-transition diagrams, and waveforms? They do not represent all types of non-textual representations.
2. It is essential to ensure that the generated error report can effectively guide the model in correcting errors. How do the authors validate its effectiveness?
3. In the "Targeted Code Repair Dataset" section, I suggest the author to  provide classification and proportion of the "minor" errors. Additionally, were any additional data augmentation measures taken for high-frequency errors during dataset construction?

---

> ### Author Response · Authors · 2024-11-19
>
> Thank you for your review. We have made modifications to our paper to address most of your concerns and uploaded a new pdf version. Please refer to our general official comments “Updates on Paper Revision” for details.
>
> We hope the following additional responses could further clarify your concerns.
>
> *Comments on Weaknesses*
>
> We propose two novel approaches to construct fine-tuning dataset for Verilog coding: mathematically rigorous correct-by-construction method to ensure solution correctness for non-textual data; and injecting common errors to open-source code for error repair dataset which show that models generalize to mitigating errors during code completion. Although our work is focused on the narrow domain of Verilog coding, we believe that our proposed methods could be generalizable (details in Appendix B), and be of value to the broad ICLR research community.
>
> We have added Appendix B, a dedicated section for further discussions and broader impacts. We have also greatly revised our manuscript especially figures and tables to improve clarity.
>
> *Q1. Why focus solely on karnaugh maps, state-transition diagrams, and waveforms? They do not represent all types of non-textual representations.*
>
> A1. As we further discussed in Appendix B1, we focused on Karnaugh maps, state-transition diagrams, and waveforms because they are widely used in hardware design and effectively capture hardware functionality, accounting for 30% of the VerilogEval-Human benchmark. While these do not cover all non-textual representations, our methods can be extended to other types, such as circuit schematics and data flow diagrams, in future work when suitable benchmarks become available.
>
> In Appendix B3 we discuss the significance of non-textual data for hardware design. These representations are widely utilized by hardware designers to mitigate the ambiguity and verbosity inherent in natural language descriptions. While they may be specific to hardware design, they are not Verilog-specific constructs and can be applied to various domain-specific languages (DSLs) for hardware design [1]. Furthermore, [2] emphasize the importance of non-textual representations, particularly visual representations, in describing hardware designs. While their work targets visual-language models and is therefore beyond the scope of this study, we recognize that similar methodologies of our work such as correct-by-construction methods could be employed to generate training data for visual representations, such as circuit schematics, data flow diagrams, and state transition graphs.
>
> *Q2. It is essential to ensure that the generated error report can effectively guide the model in correcting errors. How do the authors validate its effectiveness?*
>
> A2. The model we used for the self-consistency check, nemotron-340b-instruct, is weaker on VerilogEval than models used to generate correct/error code (CC etc. models). It is largely ineffective at correcting mistakes without proper guidance from error reports. To validate this, we prompt LLM to fix error code without error reports and obtain the fix rate of 13.3% (with error report should be 100%). The significant difference strongly emphasizes the importance of providing high-quality error reports to mitigate and fix the errors. We have updated our paper on Page 6 Line 313: “whereas directly prompting the LLM without detailed error reports could resolve only 13% of the errors”.
>
> *Q3. In the "Targeted Code Repair Dataset" section, I suggest the author provide classification and proportion of the "minor" errors. Additionally, were any additional data augmentation measures taken for high-frequency errors during dataset construction?*
>
> A3. The detailed error types of minor errors and additional information is provided in Appendix A.9. Table 15 shows the distribution of common error types in LLM-generated error reports, along with brief one-line descriptions. Most of these “minor” errors occur in solvable problems and stem from hardware-specific concepts (e.g., shift operations, timing violations) and Verilog related issues uncommon in software languages (e.g., latch hazards, casez priority conflicts). When generating targeted repair training data, we randomly sample detailed error reports and open-source code snippets, ensuring the error type distribution in training aligns with their natural occurrences.
>
> *References*
>
> [1] Batten et al, "PyHDLEval: An LLM evaluation framework for hardware design using python-embedded DSLs"
>
> [2] Chang et al, "Natural language is notenough: Benchmarking multi-modal generative AI for verilog generation"

---

> > ### Comment · Reviewer_sdjN · 2024-11-21
> >
> > Answers make sense.

---

> > > ### Author Response · Authors · 2024-11-21
> > >
> > > Thank you again for your review which have been helpful at improving our paper. We hope that our revisions and responses have effectively addressed your concerns, and we appreciate your decision to raise your rating.
> > >
> > > If you have any additional concerns or suggestions for improvement that may have impacted your decision to not award a higher rating, we would be grateful for your feedback.

---

### Official Review · Reviewer_5aNd · 2024-11-03

**Soundness:** 3
**Presentation:** 1
**Contribution:** 2
**Rating:** 6
**Confidence:** 1

**Summary:**

The paper introduces CraftRTL, a novel approach to Verilog code generation by leveraging a combination of synthetic data generation and targeted code repair to improve accuracy and robustness.

**Strengths:**

The primary contributions of this paper include the introduction of correct-by-construction data generation, which focuses on non-textual data representations that are essential for Verilog code and often challenging for LLMs. By incorporating Karnaugh maps, state-transition diagrams, and waveforms, the model’s capacity to interpret and generate these complex data formats improves. The experimental results demonstrate notable improvements over previous approaches on multiple benchmarks​.

**Weaknesses:**

The methods presented, particularly the correct-by-construction data targeting non-textual representations, are tailored heavily to Verilog-specific constructs such as Karnaugh maps, state-transition diagrams, and waveforms. While this adaptation effectively improves performance for Verilog code generation, the approach may have limited applicability to other hardware description languages or general programming languages that do not rely on these specific data formats. A broader discussion on how these techniques could be generalized would strengthen the paper's impact.

Several figures and tables in the paper, notably Figures 2, 4, 5, and 6, as well as Tables 4 and 5, suffer from presentation clarity issues. Figures lack a cohesive and clear structure, making it difficult for readers to follow the exact steps. In Tables 4 and 5, the inconsistent formatting of model types and unclear emphasis on the best-performing results within each category lead to potential confusion in understanding the experimental results.

**Questions:**

1. Could the authors discuss how this method for Verilog-specific elements might be adapted for other HDLs or general programming languages?

2. Figures 2, 4, 5, and 6, along with Tables 4 and 5, could benefit from clearer formatting and structure. Could the authors enhance these visuals to improve readability and clarify how the best results are highlighted across different model types?

---

> ### Author Response · Authors · 2024-11-19
>
> Thank you for your review. We have made modifications to our paper to address most of your concerns and uploaded a new pdf version. Please refer to our general official comments “Updates on Paper Revision” for details.
>
> We hope the following additional responses could further clarify your concerns.
>
> *Comments on Weaknesses*
>
> Thank you for raising the concerns on generalizability of our approach, suggestions of the paper to present a broader discussion, and suggestions to improve figures and tables. We have added Appendix B, a dedicated section for further discussions and broader impacts. We have also greatly revised our manuscript especially figures and tables to improve clarity.
>
>
> *Q1. Could the authors discuss how this method for Verilog-specific elements might be adapted for other HDLs or general programming languages?*
>
> A1. In Appendix B1 we discuss the generalizability of correct-by-construction methods targeting non-textual representations. We agree that our method for non-textual representation is hand-crafted and difficult to transfer. Our approach is largely inspired by [1] where symbolic deduction engines were used to generate finetuning data, improving LLM capabilities in solving Olympiad geometry problems. We hope mathematically rigorous approaches could inspire future work on improving LLMs general capabilities in areas such as math, coding, and symbolic reasoning. Moreover, we recognize that adapting these methods to other domains may require human tuning to identify the best data generation method, and we note that automating this process for scalability could be a promising future research direction.
>
> In Appendix B3 we discuss the significance of non-textual data for hardware design. These representations are widely utilized by hardware designers to mitigate the ambiguity and verbosity inherent in natural language descriptions. While they may be specific to hardware design, they are not Verilog-specific constructs and can be applied to various domain-specific languages (DSLs) for hardware design [2]. Furthermore, [3] emphasize the importance of non-textual representations, particularly visual representations, in describing hardware designs. While their work targets visual-language models and is therefore beyond the scope of this study, we recognize that similar methodologies of our work such as correct-by-construction methods could be employed to generate training data for visual representations, such as circuit schematics, data flow diagrams, and state transition graphs.
>
> Our method is inherently adaptable to other HDLs and programming languages. Leveraging custom-designed solvers to generate accurate solutions is an approach that can be applied to any programming language. While this work focuses on Verilog, it is not limited to it and can be extended to various domain-specific languages (DSLs) for hardware design. This adaptability allows the pipeline to address language-specific challenges effectively while maintaining its utility across diverse domains.
>
> *Q2. Figures 2, 4, 5, and 6, along with Tables 4 and 5, could benefit from clearer formatting and structure. Could the authors enhance these visuals to improve readability and clarify how the best results are highlighted across different model types?*
>
> A2. We appreciate the reviewer’s feedback regarding the formatting and structure of Figures 2, 4, 5, and 6, as well as Tables 4 and 5. We have thoroughly updated all the mentioned figures and tables in the revised manuscript to enhance their readability and clarity. Additionally, we have ensured that the best results across different model types are now clearly highlighted for better understanding. Thank you for bringing this to our attention. Specifically we have made the following changes:
> - Merged figures to Figure 1 and provided further details in Appendix A10.
> - Redrawn Figure 3 with abbreviated text, enlarged bolded text fonts for improved readability. Removed original Figure 2 now replaced with Figures 26,27, 28 in Appendix.
> - Redraw Figure 4 in high-resolution tikzplot.
> - Removed confusing captions and highlighted only best results for Table 4,5.
>
> *References*
>
> [1] Trinh et al, " Solving olympiad geometry without human demonstrations"
>
> [2] Batten et al, "PyHDLEval: An LLM evaluation framework for hardware design using python-embedded DSLs"
>
> [3] Chang et al, "Natural language is notenough: Benchmarking multi-modal generative AI for verilog generation"

---

> > ### Comment · Reviewer_5aNd · 2024-11-23
> >
> > Thanks for your clarification and revision. I raised my score to 6.

---

> ### Author Response · Authors · 2024-11-22
> **Gentle Reminder**
>
> Dear Reviewer 5aNd,
>
> Thank you for taking the time to review our paper, and for your thoughtful feedback. We have carefully considered your comments and submitted our revised manuscript and detailed response addressing the many valid concerns you raised.
>
> We value your expertise and input, and we would greatly appreciate any further feedback or clarification you might have regarding our response. As the discussion deadline is approaching, we wanted to kindly check if you have had the opportunity to review our revised manuscript and detailed response.
>
> Please let us know if there are any additional aspects of our work you would like us to address or elaborate on. Your guidance is invaluable in helping us improve the clarity and quality of our paper.
>
> Thank you again for your time and support.

---

> ### Author Response · Authors · 2024-11-25
>
> Thank you again for your review which have been helpful at improving our paper. We hope that our revisions and responses have effectively addressed your concerns, and we appreciate your decision to raise your rating.
>
> If you have any additional concerns or suggestions for improvement that may have impacted your decision to not award a higher rating, we would be grateful for your feedback.

---

### Official Review · Reviewer_8TDV · 2024-11-03

**Soundness:** 4
**Presentation:** 3
**Contribution:** 4
**Rating:** 8
**Confidence:** 4

**Summary:**

This paper does a thorough evaluation of LLMs for verilog code generation. They first analyze existing model performance on Verilog code generation tasks, identify that "non-textual representations" are commonly mis-reasoned about, use this to motivate two new methods for improving SDG for verilog code gen tasks, and test their approach against other SDG approaches. They find that their method outperforms baselines.

**Strengths:**

* This paper presents a clear discussion of an important and under-explored topic. Low-level programming languages are an appealing area in which to automate code reasoning, and programs in HDLs are notoriously difficult to verify.
* thorough evaluation in terms of comparison to other SDG methods and other baselines. Appropriate ablations further convey the value of all components of their approach.
* the code repair generation process is compelling, and validation well-grounded in existing literature. I anticipate that it's highly transferrable to other domains of data as well.
* the combination of using hand-crafted methods for highly-underrepresented or challenging concepts ("non-textual elements") and automated self-consistency-based methods for intermediate concepts (generating the repair data) paints a cohesive picture for SDG, especially in this domain.

**Weaknesses:**

* the data generation processes for Karnaugh maps, state-transition diagrams, and waveforms are pretty hand-crafted. This makes this method difficult to transfer to other identified model weakness categories, and requires human-tuning to identify the best data gen method per category. This approach also may not work as well, if at all, on some categories. (For example, the findings of L461 that indicate the Waveforms problems do not improve as much as the other approaches.) An automated method for designing the data construction may scale better. (out of scope for this paper though, and I would not consider this a reason for rejection)

**Questions:**

* Fig 1 is kind of confusing. Why choose checkpoints 1 and 2? Would we hope for the pass@k for checkpoint 2 to be higher than for chkpt 1?  This scatter-plot resembles a confusion matrix-- why choose the scatter plot representation over a different option? The value of figure 1 is made more apparent once we see figure 5. Maybe the two could be presented closer to one another in a camera-ready. How were the "solvable" and "unsolvable" regions chosen?
* L319: how do we know that the ability to self-correct (validating via self-consistency) is due to a good error report, and not the model's ability to correct independent of the error report? Especially since the examples from which error reports are generated did yield both correct and incorrect generations, to start with.
* is the amount of training data consistent between all rows of Table 6?

---

> ### Author Response · Authors · 2024-11-19
>
> Thank you for your review and recognition of our work. We provide the following comments regarding Weaknesses and answers to Questions.
>
> *Comments on Weaknesses:*
>
> We agree that our method for non-textual representation is hand-crafted and difficult to transfer. We also agree that automating such methods could be a promising future direction. Our approach is largely inspired by [1] where symbolic deduction engines were used to generate finetuning data, improving LLM capabilities in solving Olympiad geometry problems. We hope mathematically rigorous approaches could inspire future work on improving generic LLM capabilities. We provide such discussions on the generalizability and limitations of this approach in Appendix B1.
>
> As you highlighted, our method currently does not work well for Waveform problems. Our further analysis showed that results on sequential circuits are exceptionally poor (while combinatorial circuits near perfect). We do believe that this has to do with the quality of our manually crafted template testbench. For combinatorial circuits, we enumerate and scan through all possible inputs (2^4=16 cases in total for 4 input variables), thus our simulation is “complete”. For sequential circuits, however, we mainly rely on random test patterns (we did not consider automated test pattern generation (ATPG) methods for hardware functional coverage [2] or formal methods [3]), so we can not ensure that all states and transitions are covered. Furthermore we only presented limited simulation cycle waveforms in the problem description but still conducted full simulation for testing (similar to VerilogEval-Human benchmark). As such, the same test cases for Waveform sequential circuits could have multiple corresponding Verilog solutions (we can not guarantee one-on-one correspondence through coverage due to input delimitation). We also note that reverse engineering circuit functionality from waveforms is inherently challenging and possibly an ambiguous task.
>
> *Q1. Fig 1 is kind of confusing. Why choose checkpoints 1 and 2? Would we hope for the pass@k for checkpoint 2 to be higher than for ckpt 1? This scatter-plot resembles a confusion matrix-- why choose the scatter plot representation over a different option? The value of figure 1 is made more apparent once we see figure 5. Maybe the two could be presented closer to one another in a camera-ready. How were the "solvable" and "unsolvable" regions chosen?*
>
> A1. Thank you for the suggestion! We have updated Figure 1 and provided further information in Appendix A10.
> We choose checkpoint2 to be the last checkpoint during training and checkpoint1 to be the immediate predecessor (64 gradient steps). The ideal outcome is not merely reduced variability but also less degradations and improved accuracy: specifically, most problems in checkpoint2 should show higher pass rates than checkpoint1, assuming that training on additional data enhances model performance. We select such a representation hoping to give readers an overall impression on training variability across all problems with two checkpoints. In Table 17 of Appendix A10 we present an alternative option of displaying the pass rates for selected benchmark problems throughout the training progression. We classify problems with pass rates exceeding 67% as solvable, and those below 33% as unsolvable.
>
> *Q2. L319: how do we know that the ability to self-correct (validating via self-consistency) is due to a good error report, and not the model's ability to correct independent of the error report? Especially since the examples from which error reports are generated did yield both correct and incorrect generations, to start with.*
>
> A2. The model we used for the self-consistency check, nemotron-340b-instruct, is weaker on VerilogEval than models used to generate correct/error code (CC etc. models). It is largely ineffective at correcting mistakes without proper guidance from error reports. To validate this, we prompt LLM to fix error code without error reports and obtain the fix rate of 13.3% (with error report should be 100%). The significant difference strongly emphasizes the importance of providing high-quality error reports to mitigate and fix the errors. We have updated our paper on Page 6 Line 313: “whereas directly prompting the LLM without detailed error reports could resolve only 13% of the errors”.
>
> *Q3. Is the amount of training data consistent between all rows of Table 6?*
>
> A3. The training data size for the three models presented in Table 6 increases incrementally: SDG (80.1k), SDG-CC (108.6k), and SDG-CC-Repair (110k). We have updated Table 6 for clarification.
>
> *References*
>
> [1] Trinh et al, " Solving olympiad geometry without human demonstrations"
>
> [2] Alexander Miczo, "Digital Logic Testing and Simulation"
>
> [3] Qayyum et al, "LLM-assisted Automated Incremental Proof Generation for Hardware Verification"

---

> > ### Comment · Reviewer_8TDV · 2024-11-19
> >
> > Thank you for the clarifications. I maintain my score.

---

> ### Author Response · Authors · 2024-11-21
>
> Thank you once more for your thoughtful feedback and acknowledgment of our efforts.

---

### Author Response · Authors · 2024-11-19
**Updates on Paper Revision**

We thank all reviewers for questions and suggestions. We have made modifications to our paper to address comments from the reviewers and uploaded a new pdf version. In summary and response to all reviewers, we have made the following modifications:

1. We have greatly revised the figures and tables to enhance the visual and readability. Specifically we have:
- Merged figures to Figure 1 (Reviewer 8TDV), and provided further details in Appendix A10 (Reviewer uCA7).
- Redrawn Figure 3 with abbreviated text, enlarged bolded text fonts for improved readability (Reviewers 5aNd, sdjN). Also removed original Figure 2 now replaced with Figures 26,27, 28 in Appendix.
- Redrawn Figure 4 in high-resolution tikzplot (Reviewer 5aNd)
- Removed confusing captions and highlighted only best results for Table 4,5 (Reviewer 5aNd).

2. Added requested new results and details
- Classification and proportion of minor errors in Appendix A9 (Reviewer sdjN).
- Clarifications on effectiveness of error report in error fix on Page 6 Line 313: “whereas directly prompting the LLM without detailed error reports could resolve only 13% of the errors” (Reviewers 8TDV, sdjN, uCA7)

3. Added Appendix B for further discussions and broader impacts. In this section we address the concerns on technical contribution (Reviewer sdjN), generalizability of our proposed methods to other HDLs or programming languages (Reviewers 5aNd, uCA7), and significance on focusing to non-textual representations (Reviewer sdjN).


Although our work is focused on the narrow domain of Verilog coding, we believe that our proposed methods could be generalizable (details in Appendix B), and be of value to the broad ICLR research community. We hope our revised manuscript and rebuttal could offer clarification and hopefully resolve the many valid concerns raised by reviewers.

---

### Meta-Review · Area_Chair_Gsqk · 2024-12-19

**Metareview:**

The paper introduces methods for training code LLMs on an important hardware description language, doing a deep dive on the specific kinds of errors made on those problems and addressing them through correct-by-construction synthetic data generation. The primary strength is that it addresses an important problem and has a robust empirical evaluation, while the primary weakness is that the method is not particularly conceptually creative and relatively narrow in its practical implications. I recommend accepting this paper due to it's practical importance, and the fact that researchers in the ML and HDL communities could likely build on these models and methods.

**Additional Comments On Reviewer Discussion:**

The primary issues that came up during discussion were clarity (addressed during revision) and narrowness of the techniques/application domain (not addressed: It appears intrinsic to the work).

---

### Decision · Program_Chairs · 2025-01-22

Accept (Poster)